# Odor-evoked inhibition of olfactory sensory neurons drives olfactory perception in *Drosophila*

Li-Hui Cao[1,2,3,4], Dong Yang[1,2,3], Wei Wu[1,2,3], Xiankun Zeng[5], Bi-Yang Jing[1,2,3], Meng-Tong Li[1,2,3,6], Shanshan Qin[4], Chao Tang[3,4], Yuhai Tu[4,7] & Dong-Gen Luo[1,2,3,4]

Inhibitory response occurs throughout the nervous system, including the peripheral olfactory system. While odor-evoked excitation in peripheral olfactory cells is known to encode odor information, the molecular mechanism and functional roles of odor-evoked inhibition remain largely unknown. Here, we examined *Drosophila* olfactory sensory neurons and found that inhibitory odors triggered outward receptor currents by reducing the constitutive activities of odorant receptors, inhibiting the basal spike firing in olfactory sensory neurons. Remarkably, this odor-evoked inhibition of olfactory sensory neurons elicited by itself a full range of olfactory behaviors from attraction to avoidance, as did odor-evoked olfactory sensory neuron excitation. These results indicated that peripheral inhibition is comparable to excitation in encoding sensory signals rather than merely regulating excitation. Furthermore, we demonstrated that a bidirectional code with both odor-evoked inhibition and excitation in single olfactory sensory neurons increases the odor-coding capacity, providing a means of efficient sensory encoding.

[1] State Key Laboratory of Membrane Biology, College of Life Sciences, Peking University, Beijing 100871, China. [2] IDG-McGovern Institute for Brain Research, Peking University, Beijing 100871, China. [3] Peking-Tsinghua Center for Life Sciences, Academy for Advanced Interdisciplinary Studies, Peking University, Beijing 100871, China. [4] Center for Quantitative Biology, Academy for Advanced Interdisciplinary Studies, Peking University, Beijing 100871, China. [5] United States Army Medical Research Institute of Infectious Diseases, Frederick, MD 21702, USA. [6] Peking University-Tsinghua University-National Institute of Biological Sciences Joint Graduate Program (PTN), Peking University, Beijing 100871, China. [7] IBM T.J. Watson Research Center, Yorktown Heights, NY 10598, USA. Li-Hui Cao, Dong Yang, Wei Wu and Xiankun Zeng contributed equally to this work. Correspondence and requests for materials should be addressed to D.-G.L. (email: dgluo@pku.edu.cn)

Primary sensory receptor cells of most modalities exhibit spontaneous activities[1–6], enabling their stimulus-induced responses with an activity decrease from the baseline by sound, temperature, or chemicals. For example, in addition to exciting olfactory sensory neurons (OSNs) by increasing the action-potential firing, odors have been found to inhibit the basal firing of OSNs from insects to mammals[7–15]. While excitation is known to encode sensory information, the functional roles of inhibition in peripheral sensory receptor cells remain unsolved. Does the stimulus-evoked inhibition simply regulate the

excitability of primary sensory cells[16], or does it act as a sensory code[17]? If the latter is true, what stimulus information does it encode, and how does it improve sensory coding?

To address these questions, we examined olfaction in *Drosophila*, a well-established and genetically tractable model system[18–20]. Olfaction begins with odor detection by odorant receptors (ORs) expressed in the OSNs. In adult *Drosophila*, there are ~50 types of ORs; the OSNs expressing a given OR converge their axons to one of ~50 glomeruli in the antennal lobe. This olfactory architecture is conserved from insects to mammals,

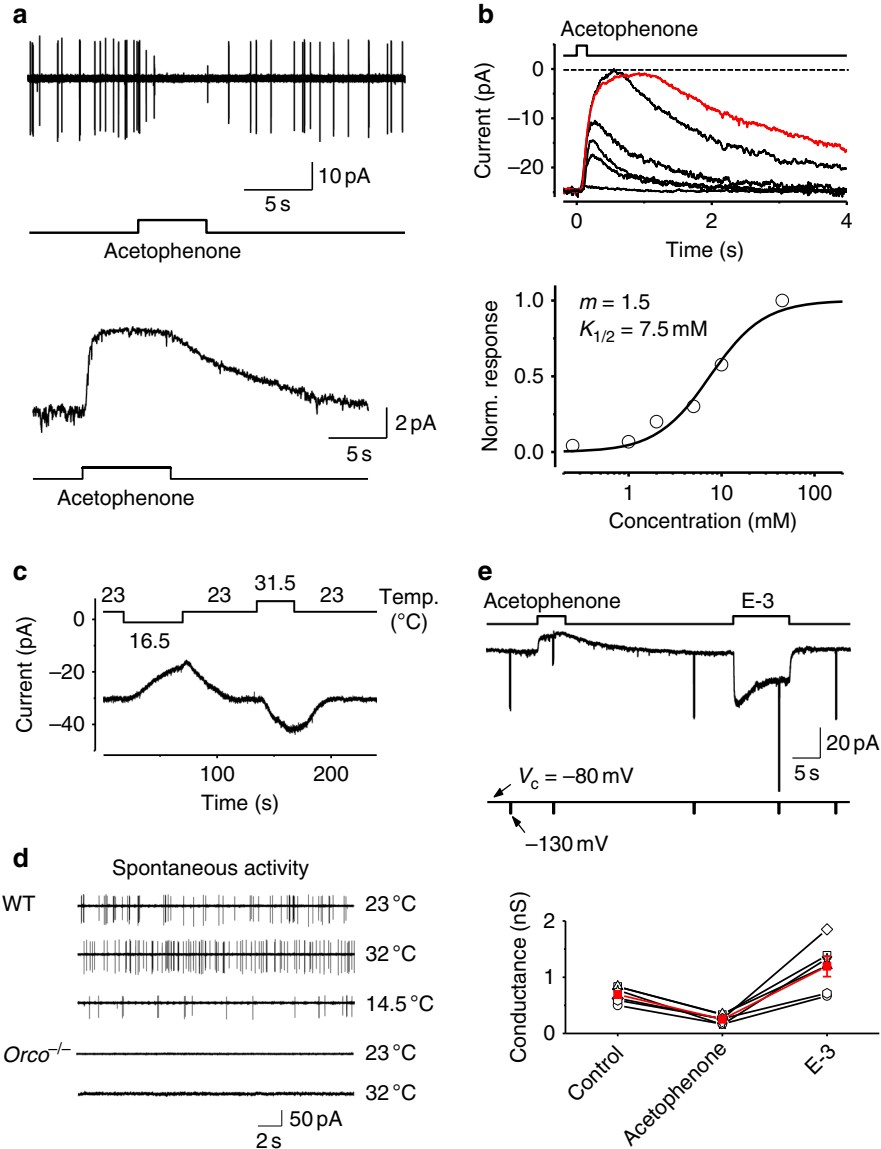

**Fig. 1** Outward receptor currents induced by inhibitory odors in Or85a-expressing OSNs. **a** Acetophenone (10 mM) abolishes the spontaneous firing (cell-attached recording, top) and triggers an outward receptor current (perforated patch-clamp recording, voltage-clamped at −80 mV, bottom). The timing of odor application is indicated. **b** Dose–response relationship of the odor-evoked inhibition. Top, superimposed traces of responses to 150-ms pulses of acetophenone at 0.25, 1.25, 5, 10, and 45 mM. The inhibitory responses to acetophenone at 45 mM (maximal water solubility) with durations of 150 and 500 ms (red trace) exhibit the same peak amplitude and reduce the basal inward current to 0 pA. Each trace is the average of 5–10 trials. Bottom, the normalized dose–response relationship. The fit is the Hill equation, $R/R_{max} = C^m / (C^m + K_{1/2}{}^m)$, where $R$ is the peak-response amplitude, $R_{max}$ is the saturated peak response, $C$ is the odor concentration, $K_{1/2}$ is the odor concentration that half-saturates the response, and $m$ is the Hill coefficient. In this experiment, $K_{1/2} = 7.5$ mM and $m = 1.5$. Collective results ($n = 4$): $K_{1/2} = 27 \pm 13$ mM, and $m = 1.4 \pm 0.2$. **c** Temperature dependence of the basal inward current in WT flies. **d** Temperature dependence of spontaneous firing in WT flies (three top traces) and $Orco^{-/-}$ flies (two bottom traces). **e** Odor-evoked inhibition decreases membrane conductance. Top, membrane conductance was monitored using 10-ms voltage pulses (stepping from the holding voltage of −80 mV to −130 mV) before, during, and after the application of acetophenone (10 mM) and ethyl 3-hydroxybutyrate (E-3) (1 mM). The timing of odor stimulation and voltage pulses is indicated above and below the response trace, respectively. Bottom, collective data of membrane-conductance changes, with average indicated in red

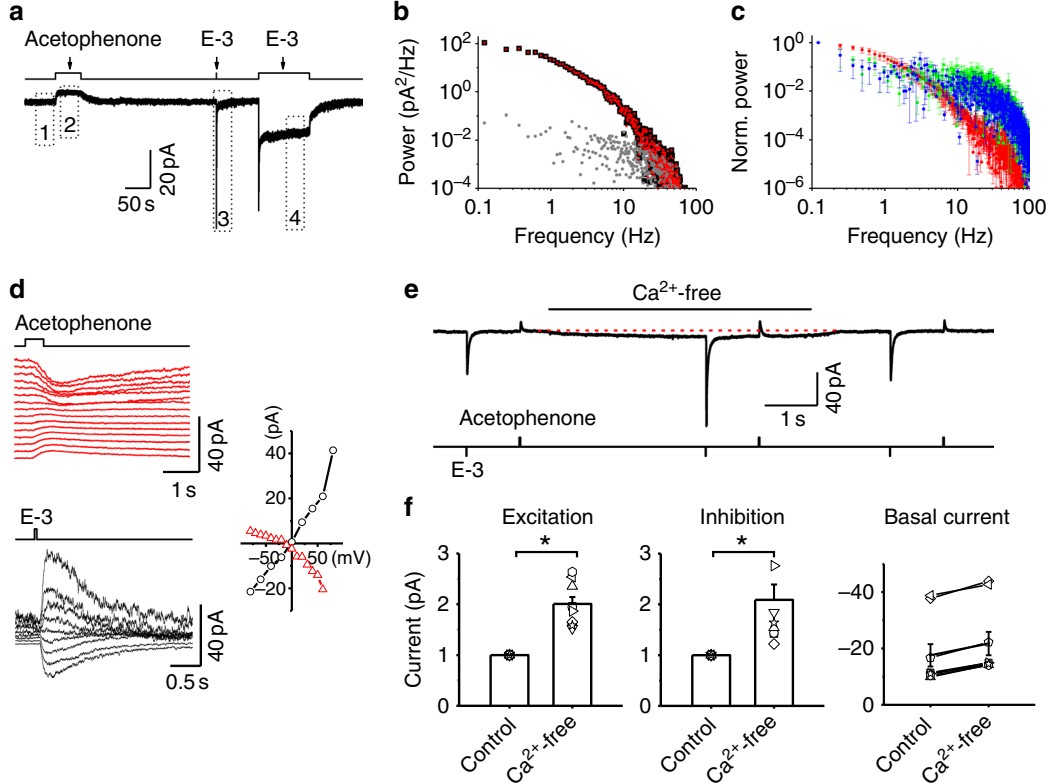

**Fig. 2** Signaling by spontaneously and odor-activated ORs. **a** The basal inward current and odor-evoked receptor currents to acetophenone (20 mM; 30 s) and E-3 (1 mM; 35 ms and 100 s). **b** Power spectrum of excitatory responses. Gray and black represent the power spectra of segments 1 and 3 in **a**, respectively; red represents the difference spectrum of segments 3−1. **c** Scaled power spectra of the basal activities (segments 1−2, blue) and the excitatory response to a pulse (segments 3−1, red), and a step (segments 4−1, green) of E-3. $N = 3$, error bars represent SEM. **d** I–V relationships. Voltage dependence of the receptor currents induced by inhibitory (left, top) and excitatory (left, bottom) odors. Current–voltage relationships of inhibitory and excitatory responses (right). The reversal potentials of inhibitory and excitatory responses are $-5.3 \pm 2.0$ and $-2.5 \pm 2.0$ mV ($n = 5$), respectively. Acetophenone: 20 mM; E-3: 2 mM. **e** Calcium modulation of the basal inward current and odor responses. Removal of extracellular calcium increases the basal current, outward receptor current to acetophenone (20 mM, 150 ms), and inward receptor current to E-3 (1 mM, 35 ms). **f** Removal of extracellular calcium increases the excitatory (left), inhibitory (middle) responses, and basal current (right). $N = 9$, error bars represent SEM, *$P < 0.05$

suggesting a common solution to olfaction[18]. Odor-evoked inhibition exists in most *Drosophila* OSNs[15, 21, 22], providing an opportunity to investigate its molecular origin, physiological functions, and computational roles in shaping odor perception.

By combining molecular genetics, electrophysiology, two-photon calcium imaging, optogenetics, behavioral studies and computations, we report for the first time that odor-evoked inhibition of *Drosophila* OSNs directly encodes odor identity and drives both attraction and avoidance behaviors. A single type of OSNs with odor-evoked inhibition and activation can drive two opposing behaviors and can also effectively discriminate odor mixtures. Notably, the blockage of synaptic transmission of odor-evoked inhibition can result in a complete switch of olfactory behaviors. Mechanistically, such inhibition is caused by a direct odor inhibition of the constitutively activated ORs. A bidirectional odor response with both odor-evoked inhibition and activation in the same OSNs increases odor-coding capacity by reducing response saturation and decorrelating odor representation. Taken together, our work demonstrates that odor-evoked inhibition of OSNs is comparable to odor-evoked activation in encoding odor information for behavior and perception in *Drosophila*.

## Results
### An outward receptor current underlies odor-evoked inhibition.
In *Drosophila* OSNs, both the levels of basal activities and the

modes of odor-evoked responses (activation vs. inhibition) are determined by the expressed ORs[21, 22]. However, a mechanistic understanding of the connections between these two phenomena has been hampered by difficulties in recording intracellularly from OSNs. We used our recent technical advances[23] to perform patch-clamp recordings on *Drosophila* OSNs and investigate the molecular origin of odor-evoked inhibition. The *Or85a*-expressing OSNs exhibited spontaneous firing in the absence of stimuli, and this effect was reversibly abolished by acetophenone (Fig. 1a, top). Correspondingly, an outward receptor current was induced by acetophenone under a voltage-clamped configuration (Fig. 1a, bottom). This acetophenone-induced inhibition was a direct effect on *Or85a*-OSNs rather than an ephaptic inhibition[24] because similar results were obtained in isolated *Or85a*-OSNs (Supplementary Fig. 1a). This conclusion was further supported by a similar inhibition in *Or85a*-OSNs of *Orco*−/− flies[25] with *Orco* restored only to *Or85a*-OSNs (Supplementary Fig. 1b).

In the absence of odor stimuli, we recorded a basal inward current of $-18.2 \pm 14.4$ pA ($n = 42$, mean $\pm$ SD) in *Or85a*-OSNs at 23 °C, suggesting the existence of ion channels that are constitutively open. Acetophenone evoked outward receptor currents in a dose-dependent manner (Fig. 1b and Supplementary Table 1), with the same amplitude of its maximal responses as the basal inward current. These results imply that the acetophenone-induced outward receptor current is a reduction of the basal inward current. We found that the basal inward current was

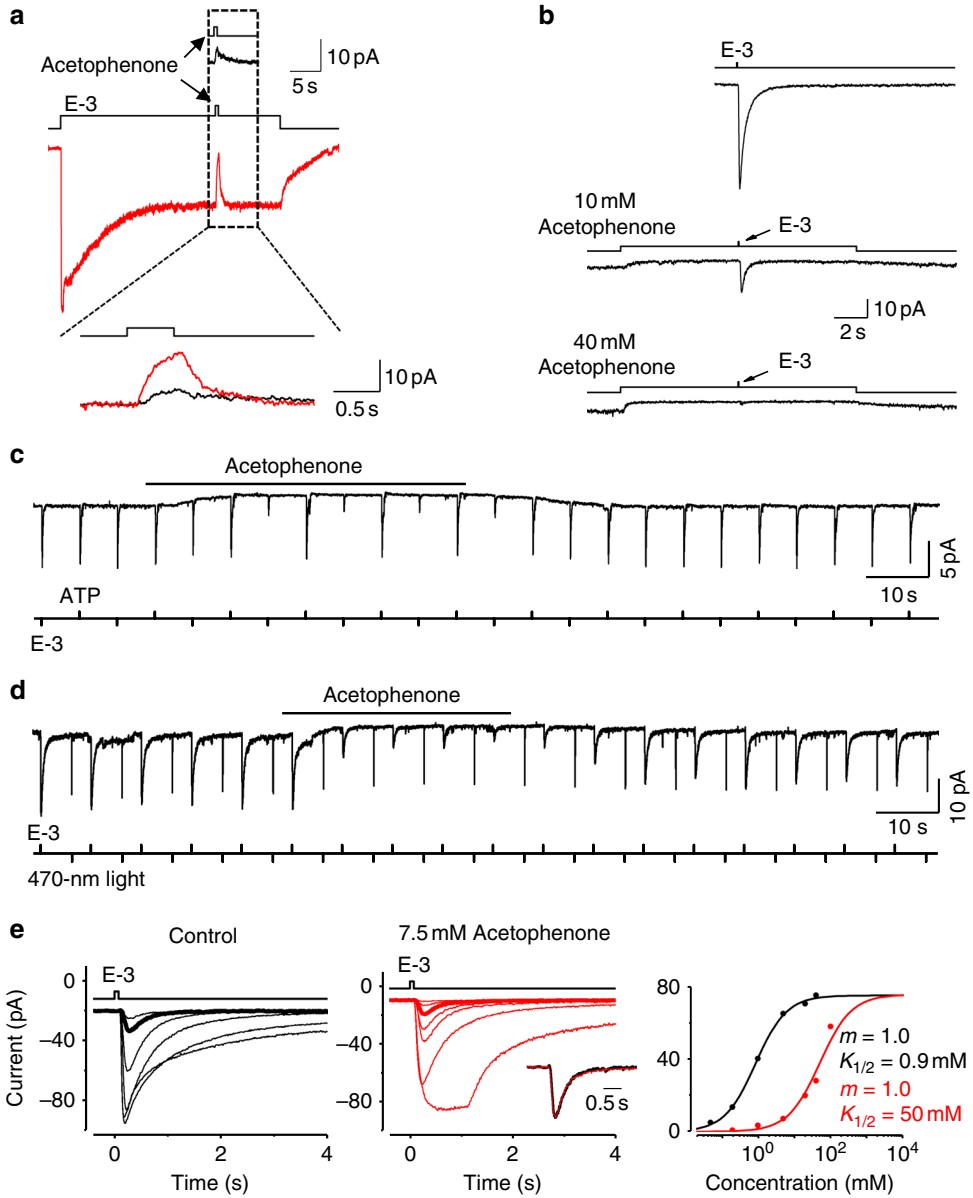

**Fig. 3** Interactions between odor-evoked inhibition and activation in the same OSNs. **a** Acetophenone reduces the basal current and odor-evoked inward current. Top, the acetophenone (10 mM)-induced response; middle, the response induced by acetophenone (10 mM) in the presence of E-3 (4 mM); bottom, the overlay of acetophenone-induced responses with (red) and without (black) the presence of E-3. E-3 increases acetophenone-induced responses from $3.0 \pm 0.9$ to $15 \pm 2$ pA (mean $\pm$ SEM; $n = 6$; $P < 0.001$). **b** Dose-dependence of acetophenone-induced reduction of excitatory responses. The inward receptor current induced by E-3 (4 mM, 35 ms) (top) is inhibited by background application of 10 mM (middle) or 40 mM (bottom) acetophenone. Acetophenone decreases E-3-induced responses from $51.3 \pm 16.0$ pA (control, $n = 5$) to $23.0 \pm 8.0$ pA (10 mM acetophenone) to $6.0 \pm 4.0$ pA (40 mM acetophenone). **c** Acetophenone does not inhibit $P2X_2$-mediated responses. The Or85a-OSNs express exogenous ATP-gated $P2X_2$ cation channels. The inward current triggered by 1 mM E-3 is decreased to $23 \pm 3\%$ ($n = 22$) by 5 mM acetophenone, but the inward current triggered by 1 mM ATP is unaffected by acetophenone. Note the reduction of the basal inward current by acetophenone. **d** Acetophenone does not inhibit ChR2-mediated inward current, but decreased the E-3-induced responses to $13 \pm 3\%$ ($n = 32$). Acetophenone: 5 mM; E-3: 1 mM, 35 ms, light: 10 ms. **e** Competitive inhibition of the excitatory responses by acetophenone. Superimposed traces showing responses to E-3 (35 ms) in the absence (black, left) and presence (red, middle) of 7.5 mM acetophenone. The red trace of the largest amplitude was obtained with a 1-s pulse of 70 mM E-3. The normalized responses of the two thicker traces with similar small response amplitudes have identical response shape and kinetics (inset, middle). Dose–response relationships (right) for the data with and without the presence of acetophenone are shown in red and black, respectively. 7.5 mM acetophenone decreases the sensitivity to E-3 from a $K_{1/2}$ of $1.1 \pm 0.2$ to $40 \pm 10$ mM ($n = 3$; $P < 0.001$)

completely eliminated after ablation of *Orco* (Supplementary Fig. 1c, d), indicating its origin from the constitutive activity of OR/ORCO complex. We further found that the basal inward current was temperature-dependent, increasing at higher temperatures and decreasing at lower temperatures (Fig. 1c and Supplementary Fig. 1e). Similarly, the spontaneous firing in

Or85a-OSNs also depended on temperature and the presence of *Orco* (Fig. 1d and Supplementary Fig. 1f, g), suggesting that the spontaneous firing in Or85a-OSNs was mainly driven by the basal activity of ORs.

The outward receptor current that underlies acetophenone-evoked inhibition could be produced by distinct mechanisms.

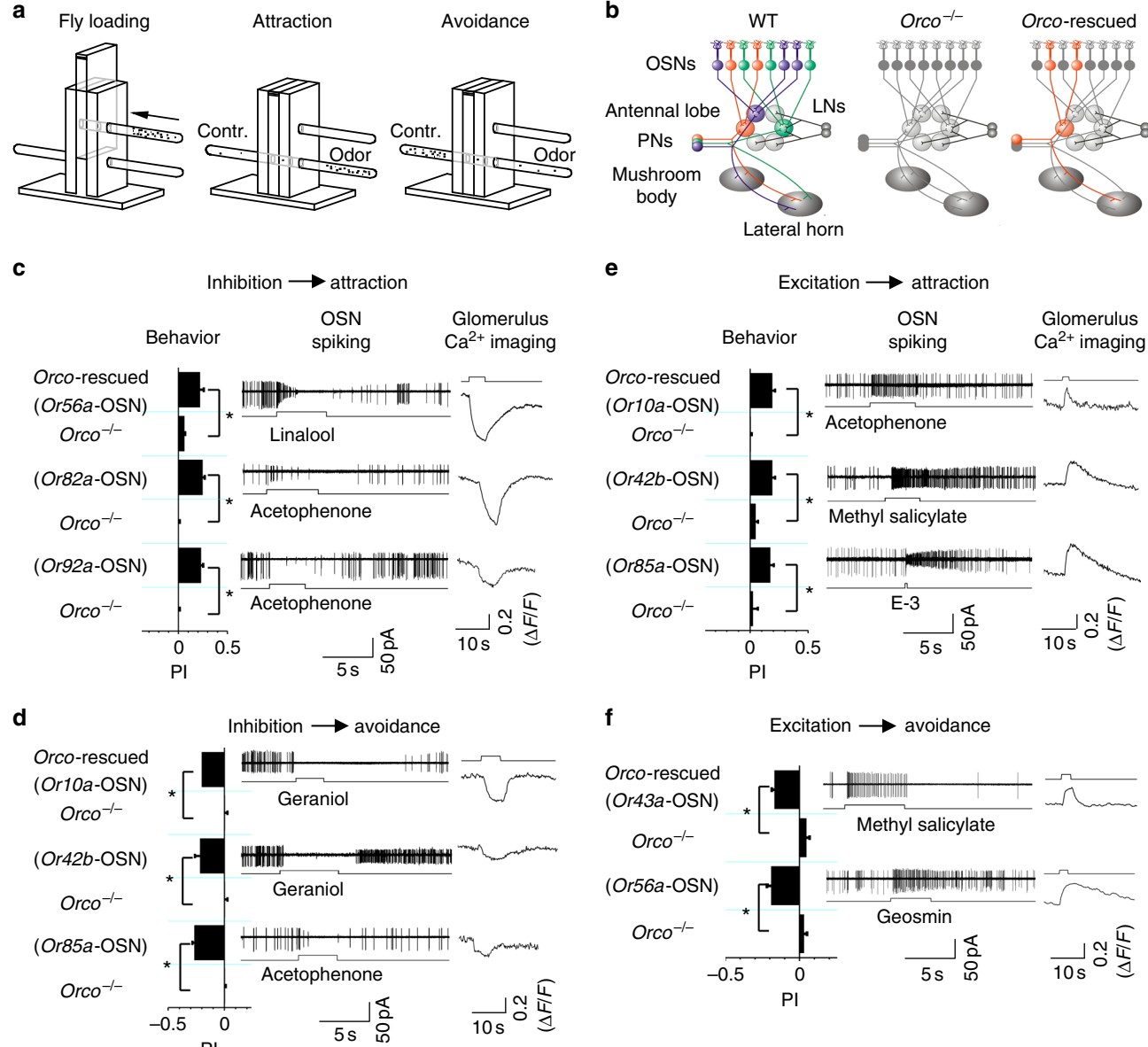

**Fig. 4** Odor-evoked OSN inhibition and olfactory behaviors. **a** Schematic T-maze behavioral assay. Fly loading to the T-maze (left); attraction by odors (middle); repellence by odors (right). **b** Olfactory neural pathways in *Drosophila*. There are ~50 types of ORs in the adult antennae, with OSNs expressing the same type of OR (marked by one color) converging onto a glomerulus in the antennal lobe (left). In *Orco*[−/−] flies (middle), the ORs are not functional because of their dependence on *Orco*[25]; in *Orco*-rescued flies (right), *Orco* is restored to the OSNs expressing a specific type of OR (marked in orange) in an *Orco*[−/−] background, thus restoring the odor-sensing ability in the targeted OSNs. **c** Odor-evoked inhibition elicits attraction. Odors inhibiting the basal firing of OSNs (middle) and calcium signals in corresponding glomeruli (right) attract flies (left) with *Orco* restored to OSNs expressing *Or56a*, *Or82a*, or *Or92a*, as indicated. **d** Odor-evoked inhibition elicits avoidance. **e** Odor-evoked activation elicits attraction. **f** Odor-evoked activation elicits avoidance. Odor concentrations: acetophenone, linalool, and methyl salicylate at a dilution of 10[−3] (vol/vol); geraniol at 10[−4]; E-3 at 10[−2]; and geosmin at 10[−5]. $N = 8–20$, error bars represent SEM, *$P < 0.01$. LNs local neurons, PNs projection neurons

One possibility is that acetophenone binds to *Or85a* receptor proteins to open ion channels with a reversal potential below the resting potential of OSNs, exemplified by the opening of potassium channels that mediate odor-evoked inhibition in lobster and toad OSNs[11, 26]. Alternatively, acetophenone may interact with *Or85a* receptors to close ion channels that are constitutively open and have a reversal potential above the resting potential of OSNs. To differentiate between the two possibilities, we measured changes in membrane conductance to determine whether the odor-evoked inhibition was caused by the opening or closing of ion channels[10]. We found that acetophenone dramatically decreased the membrane conductance when probed

with pulses of hyperpolarizing voltage (Fig. 1e), thus demonstrating the closure of ion channels that are constitutively open. In contrast, ethyl 3-hydroxybutyrate, an excitatory odor for *Or85a*-OSNs, increased the membrane conductance of the same OSNs and produced an inward receptor current (Fig. 1e), indicating the opening of ion channels.

Together, our results demonstrated that odor-evoked OSN inhibition is produced by the closure of ion channels that are opened by constitutively/thermally activated ORs, thus yielding an outward receptor current. Our findings support the hypothesis that OR molecules fluctuate between active and inactive states in a temperature-dependent manner, with inhibitory odors

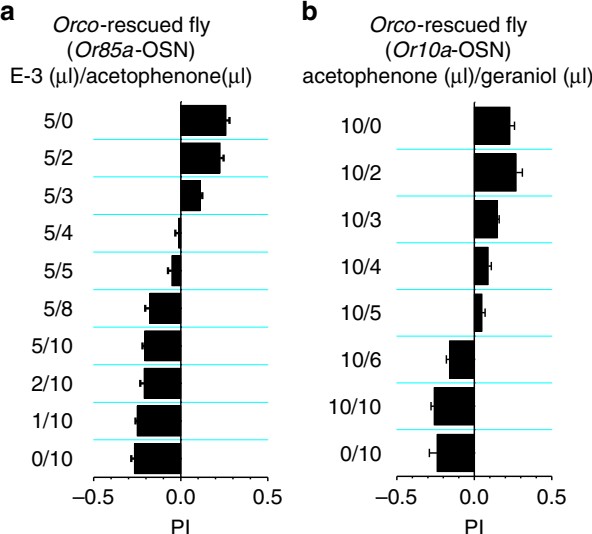

**Fig. 5** Discriminating odor mixtures by a single type of OSN. **a** Behaviors of the flies with *Orco* restored to *Or85a*-OSNs to mixtures of E-3 (increasing firing) and acetophenone (reducing basal firing) at different ratios, as indicated. **b** Behaviors of the flies with *Orco* restored to *Or10a*-OSNs to mixtures of acetophenone (increasing firing) and geraniol (reducing basal firing) at different ratios. $N = 8$–12, error bars represent SEM

stabilizing their inactive states[21]. However, our results differ from previous findings in other species in which odor-evoked inhibition of OSNs is mediated via the opening of potassium channels[11, 26].

**Signaling by constitutively or odor-activated ORs.** To investigate the signaling properties of constitutively activated ORs, we recorded the basal inward current and receptor currents evoked in *Or85a*-OSNs by their inhibitory odor acetophenone and excitatory odor ethyl 3-hydroxybutyrate (Fig. 2a). Power spectral analysis revealed similar waveforms between the basal inward current and odor-evoked responses (Fig. 2b, c), indicating that constitutively activated ORs and odor-activated ORs have similar signaling dynamics.

The finding that acetophenone-evoked inhibitory responses are a direct reduction of the basal inward current allows us to study the basal inward current by examining the inhibitory responses. Next, we examined the current–voltage relationship of acetophenone-evoked inhibitory responses and ethyl 3-hydroxybutyrate-evoked excitatory responses in the same *Or85a*-OSN. We found that inhibitory responses exhibited similar, although opposite in direction, current–voltage relationship with an identical reversal potential (Fig. 2d), indicating that the odor-evoked inhibitory and excitatory responses were mediated by similar or even identical ion channels.

Previously, we have showed that odor-evoked excitatory responses in *Drosophila* OSNs are modulated by calcium[23]. We reasoned that the basal inward current and odor-evoked inhibitory responses would show similar modulatory effects to calcium, if they share similar signaling pathways with the excitatory responses. Consistent with our prior findings[23], the removal of extracellular calcium increased the ethyl 3-hydroxybutyrate-evoked excitatory responses in *Or85a*-OSNs (Fig. 2e, f). At the same time, the removal of extracellular calcium also increased the inhibitory odor responses and basal inward current (Fig. 2e, f). These results are consistent with a model where constitutively activated ORs are likely to share a similar signaling mechanism with odor-activated ORs in

generating inward receptor currents, although some caveats about the interpretation remain.

**Interaction between odor-evoked inhibition and activation.** The coexistence of odor-evoked inhibitory and excitatory responses in the same OSN raises a possibility that the two responses may interact with each other. Next, we examined whether acetophenone could inhibit the ethyl 3-hydroxybutyrate-induced excitatory responses in *Or85a*-OSNs. In the presence of a background ethyl 3-hydroxybutyrate, a larger acetophenone-induced outward receptor current was obtained (Fig. 3a), suggesting that acetophenone could inhibit the basal inward current and odor-evoked excitatory responses. Acetophenone inhibited the inward receptor currents triggered by ethyl 3-hydroxybutyrate in a dose-dependent manner (Fig. 3b), but it did not inhibit the inward currents mediated, for example, by the exogenously expressed ATP-gated $P2X_2$ cation channels (Fig. 3c and Supplementary Fig. 2a, see also "Methods" section) or by the light-gated channelrhodopsin ChR2 (Fig. 3d and Supplementary Fig. 2b, c, see also "Methods" section). Therefore, in *Or85a*-OSNs, acetophenone specifically inhibited the activity of *Or85a* receptors.

To further explore the property of acetophenone-induced inhibition, we examined the dose–response relationship of the excitatory responses to ethyl 3-hydroxybutyrate in the presence of a background acetophenone. We found that acetophenone shifted the dose–response relationship of excitatory responses by significantly increasing the half-saturating odor concentration (Fig. 3e), while maintaining the amplitude of the maximal excitatory responses and the kinetics and shape of the responses (Fig. 3e and Supplementary Table 1). These results indicate a competitive inhibition of ethyl 3-hydroxybutyrate-induced excitatory responses by acetophenone.

**OSN inhibition independently drives olfactory behaviors.** To investigate the physiological functions of odor-evoked inhibition in the OSNs, we examined whether this inhibition by itself could elicit olfactory behaviors (Fig. 4a). A single odor simultaneously excites and inhibits different OSNs[21, 22]. To exclude the confounding effects from activation of OSNs expressing other ORs (Fig. 4b), we generated different flies in which *Orco* was restored to a single type of OSNs that expressed *Or10a*, *Or42b*, *Or43a*, *Or49b*, *Or56a*, *Or82a*, *Or85a*, or *Or92a* in an $Orco^{-/-}$ background[27]. To further limit contributions from the *Orco*-independent antennal chemosensitive neurons expressing either ionotropic or gustatory receptors[28–31], we focused on odors that did not evoke chemotaxis in flies of an $Orco^{-/-}$ background (Supplementary Fig. 3).

We used cell-attached recordings to identify odors that inhibited the basal firing of the *Orco*-restored OSNs in the flies generated above (Fig. 4c, d). The odor-evoked inhibition was further confirmed in the glomeruli by using two-photon or confocal imaging of GCaMP6m fluorescence expressed in the *Orco*-restored OSNs (Fig. 4c, d and Supplementary Fig. 4).

Unexpectedly, linalool, an odor that inhibits the basal firing and calcium signals of *Or56a*-OSNs and that does not elicit chemotaxis in $Orco^{-/-}$ flies, attracted flies with *Orco* restored to *Or56a*-OSNs (Fig. 4c). Similar inhibition-elicited attraction behaviors were observed in flies with *Orco* restored to either *Or82a*-OSNs or *Or92a*-OSNs (Fig. 4c). In contrast, geraniol, an odor that inhibits *Or10a*-OSNs, repelled flies with *Orco* restored to *Or10a*-OSNs (Fig. 4d). Similar avoidance behaviors elicited by inhibition were also observed in flies with *Orco* restored to *Or42b*-OSNs or *Or85a*-OSNs (Fig. 4d). On the other hand, odor-evoked activation alone also elicited attraction (Fig. 4e) and avoidance (Fig. 4f) behaviors depending on the type of OSNs activated.

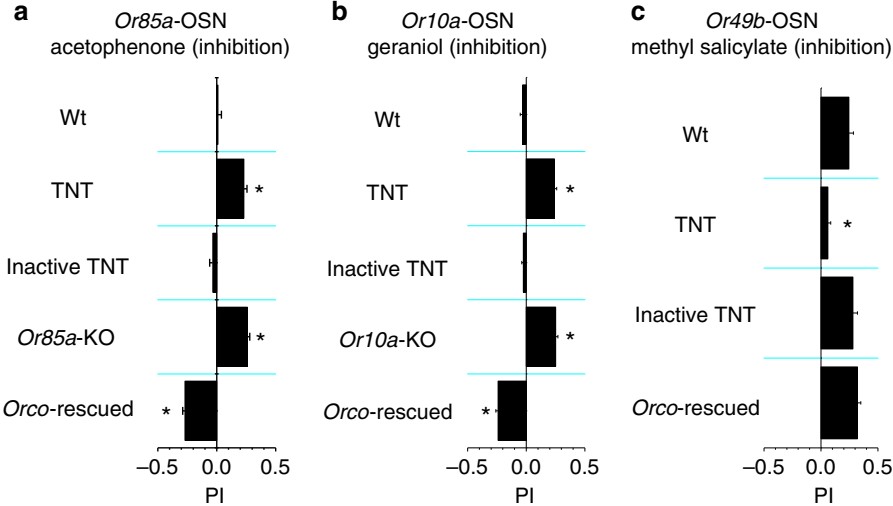

**Fig. 6** Odor coding by odor-evoked OSN inhibition in WT flies. **a** Acetophenone-elicited behaviors in WT flies, flies expressing *TNT* in *Or85a*-OSNs, flies expressing inactive *TNT* in *Or85a*-OSNs, mutant flies of knocking out *Or85a*, and flies with *Orco* restored to *Or85a*-OSNs in an *Orco*$^{-/-}$ background. **b** Geraniol-elicited behaviors in WT flies, flies expressing *TNT* in *Or10a*-OSNs, flies expressing inactive *TNT* in *Or10a*-OSNs, mutant flies of knocking out *Or10a*, and flies with *Orco* restored to *Or10a*-OSNs in an *Orco*$^{-/-}$ background. **c** Methyl salicylate-elicited behaviors in WT flies, flies expressing *TNT* in *Or49b*-OSNs, flies expressing inactive *TNT* in *Or49a*-OSNs, and flies with *Orco* restored to *Or49b*-OSNs in an *Orco*$^{-/-}$ background. *N* = 8–12, error bars represent SEM

These gain-of-function results demonstrated that similar to odor-evoked activation of OSNs, odor-evoked inhibition of basal activities in the OSNs also encodes odor information for perception and behaviors.

**Discriminating odor mixtures by using a single type of OSN**. The above results also show that a single type of OSNs could drive two opposing behaviors when inhibition and activation coexist. For example, in flies with *Orco* restored to *Or85a*-OSNs, the inhibitory and excitatory odors elicited opposing behaviors, that is, avoidance in response to acetophenone (Fig. 4d) and attraction to ethyl 3-hydroxybutyrate (Fig. 4e). These results indicate that flies with *Orco* restored to *Or85a*-OSNs may have an ability to discriminate among the mixtures of inhibitory and excitatory odors. By examining chemotaxis of these flies to such mixtures, we found that a full range of behaviors from attraction to avoidance were elicited by mixtures at different ratios (Fig. 5a). A strong attraction was elicited at a high ratio of ethyl 3-hydroxybutyrate to acetophenone (Fig. 5a). However, attraction gradually decreased as the ratio was lowered, and avoidance was eventually elicited (Fig. 5a).

Similarly, flies with *Orco* restored to the *Or10a*-OSNs were also able to discriminate mixtures of acetophenone (activating *Or10a*-OSNs and eliciting attraction behaviors, Fig. 4e) and geraniol (inhibiting *Or10a*-OSNs and eliciting avoidance behaviors, Fig. 4d). A strong attraction was elicited at a high ratio of acetophenone to geraniol (Fig. 5b). Attraction gradually decreased as the ratio was lowered, and avoidance was eventually elicited (Fig. 5b). Our spike recordings on *Or10a*-OSNs in these transgenic flies revealed a gradual transition from an increase of spike firing to a decrease of spike firing relative to the baseline level when the mixture ratio was lowered (Supplementary Fig. 5).

Therefore, we demonstrate that odor-evoked bidirectional responses in the same OSN enable olfactory computations at the level of single OSNs, which can be used to effectively discriminate odor mixtures.

**Inhibition contributes to odor coding in wild-type flies**. In adult *Drosophila*, odor recognition is based on the activity

patterns of ~50 ORs. Our gain-of-function studies have revealed that odor-evoked OSN inhibition encodes odor information in flies that have only one type of functional OR. A question is whether such inhibition could contribute to odor coding when many ORs are functional. To address this question, we expressed tetanus toxin (TNT) in *Or85a*-OSNs to block their synaptic transmission to the antennal lobe (see "Methods" section). We examined the chemotactic behaviors of these flies to acetophenone that inhibits the basal firing of Or85a-OSNs, and found that *Or85a*-TNT flies were attracted to acetophenone, which normally did not elicit chemotaxis in wild-type (WT) flies (Fig. 6a). In contrast, flies expressing an inactive TNT exhibited no chemotaxis to acetophenone (Fig. 6a). Knockout of *Or85a* receptor elicited attraction behaviors to acetophenone (Fig. 6a, see also "Methods" section). These loss-of-function results indicate that olfactory system integrates an avoidance signal from Or85a and an attraction signal collectively from other chemoreceptors, thereby leading to a non-chemotactic behavior to acetophenone in WT flies.

Similar results were obtained by disrupting the signaling of odor-evoked inhibition in *Or10a*-OSNs. Both WT flies and transgenic flies with inactive TNT expressed in *Or10a*-OSNs exhibited no chemotaxis to geraniol (Fig. 6b), an inhibitory odor for *Or10a*-OSNs. In contrast, *Or10a*-TNT flies were attracted to geraniol (Fig. 6b). Knockout of *Or10a* receptor also elicited attraction behaviors to geraniol (Fig. 6b, see also "Methods" section). Thus, geraniol-evoked inhibition in *Or10a*-OSNs contributes to the integration of geraniol-elicited behaviors in WT flies.

To test the generality of the above findings, we also examined flies with a disruption of odor-evoked inhibition in *Or49b*-OSNs. Methyl salicylate, an inhibitory odor to *Or49b*-OSNs (Supplementary Fig. 6), elicited attraction in WT flies (Fig. 6c), but the attraction was lost in transgenic flies with TNT expressed in *Or49b*-OSNs (Fig. 6c). These results imply that methyl salicylate-evoked inhibition in *Or49b*-OSNs is a dominant drive for the methyl salicylate-elicited attraction in WT flies. This conclusion is supported by our finding that *Orco*$^{-/-}$ flies with *Orco* restored to *Or49b*-OSNs was attracted by methyl salicylate (Fig. 6c).

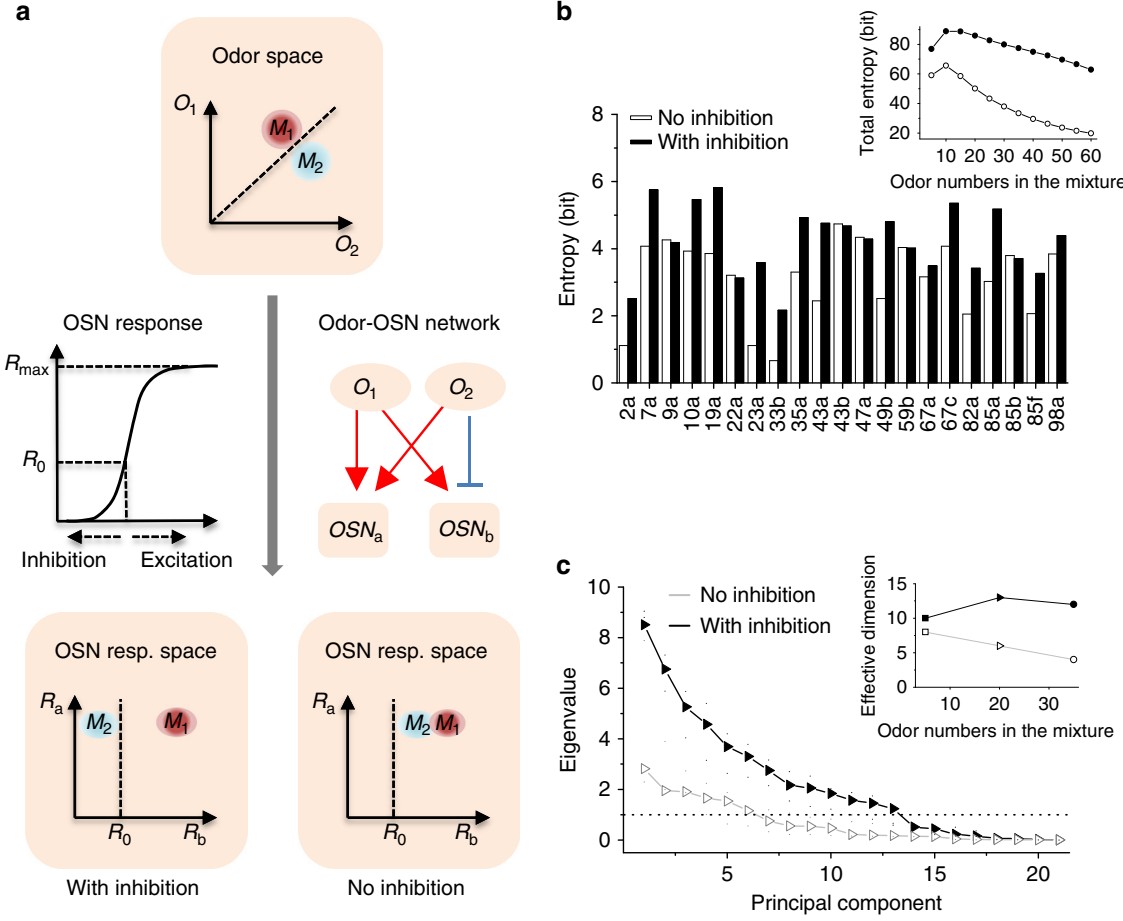

**Fig. 7** Bidirectional odor coding by OSNs. **a** Schematic odor representation by OSNs. Two similar odor mixtures, $M_1$ and $M_2$, are composed of two odors, $O_1$ and $O_2$ (top). The dashed line corresponds to equal concentrations of $O_1$ and $O_2$. After signal transformation, the two mixtures are represented in the OSN response space (bottom). The first transformation is achieved by the OSN response properties (left, middle), including response modes and sensitivity. The second transformation is effected by the odor-OSN network, with red arrows representing activation and blue representing inhibition (right, middle). $OSN_a$ is excited by $O_1$ and $O_2$; $OSN_b$ is excited by $O_1$ but inhibited by $O_2$. The responses of $OSN_a$ to each of the two mixtures likely approach saturation after summing the excitatory responses to individual odors, and $M_1$ and $M_2$ are not well separated by $OSN_a$. In contrast, $M_1$ and $M_2$ are better separated by $OSN_b$ because the inclusion of inhibition yields a dominant inhibition for $M_2$ (with the concentration of inhibitory odor $O_2$ being higher than excitatory odor $O_1$), but a dominant activation for $M_1$ (left, bottom). In the absence of inhibition, the separation between $M_1$ and $M_2$ is smaller (right, bottom). $R_{max}$ and $R_0$ are the maximal responses and the basal activity of OSNs, respectively; $R_a$ and $R_b$ are odor-evoked responses of $OSN_a$ and $OSN_b$, respectively. **b** Odor-evoked inhibition increases the capacity of odor encoding by OSNs. The coding capacity quantified by the entropy of individual OR/OSNs is computed (see "Methods" section) based on the OR-response matrix[24]. The inclusion of inhibition increases the entropy in most OSNs for mixtures containing 10 odors. The inset is the total entropy summed over all ORs/OSNs responding to mixtures of varying odor numbers. **c** Odor-evoked inhibition decorrelates odor representation. Principal component analysis of the OSN responses to 20-odor mixtures (see "Methods" section) based on the OR-response matrix[24] showed that the inclusion of inhibition increases eigenvalues of principal components and decorrelates odor representation. Consequently, as shown in the inset, the inclusion of inhibition increases the number of effective coding dimensions that have eigenvalues above a noise threshold (dotted line)

Taken together, these results imply that the odor-evoked inhibition of basal activities in OSNs plays an important role in odor coding in WT flies. However, one caveat of these experiments is that, in addition to disrupting odor-evoked OSN inhibition, these experimental manipulations also eliminate the basal firing of OSNs, which may affect the sensitivity of the downstream olfactory neurons[3].

**Inhibition increases the capacity of odor coding**. The above gain-of-function and loss-of-function studies reveal that odor-evoked OSN inhibition is comparable to odor-evoked activation as a primary odor code. Therefore, the combinatorial odor coding in the olfactory periphery is based on not only excitatory[27, 32–35] but also inhibitory responses across the OSNs. To explore the

effects of this bidirectional odor coding scheme, we developed a computational model based on the measured odor-OSN response matrix[22] to investigate whether and how the implementation of both odor-evoked inhibition and activation in single OSNs improves odor coding.

Suppose that there are $N$ odors in a mixture with the concentration of odor $i$ (=1, 2, 3…, $N$) given by $C_i$ and there are $M$ OSNs with odor-evoked responses of OSN $j$ (=1, 2, 3…, $M$) denoted by $R_j(\vec{C})$, where $\vec{C} = (C_1, C_2, C_3, \dots, C_N)$ is the $N$-dimensional concentration vector. To determine the odor-evoked response $R_j(\vec{C})$, we developed a simple model in which ORs have two states: an inactive state that produces no OSN activity and an active state that produces a maximum OSN activity $R_{max}$. Odor $i$ binds to OSN $j$ with a dissociation constant $K_{ij}$ and modulates the transition between the two OR states: excitatory odors stabilize

the active states and inhibitory odors stabilizes the inactive states. As explained in detail in the "Methods" section, we obtain

$$R_j = R_{\max}\left[1 + \alpha_j \prod_{i=1}^{N}\left(1 + \frac{C_i}{K_{ij}}\right)^{-w_{ij}}\right]^{-1} \quad (1)$$

where $w_{ij} = \{1, -1, 0\}$ for excitatory, inhibitory, and null responses, respectively. The parameter $\alpha_j$ determines a basal activity $R_{j0} = R_{\max}/(1 + \alpha_j)$. Here, we assume that different odor molecules bind with different sites on the OR receptor. However, similar results were obtained for odor molecules competing for the same binding site (see "Methods" section for details).

The transformation from an odor mixture characterized by the concentration vector $\vec{C}$ to the OSN response vector $\vec{R} = (R_1, R_2, R_3, \dots, R_M)$ determines how odors are represented (coded) by the OSNs (Fig. 7a). As shown explicitly in Eq. (1), the odor coding scheme is specified by the odor–OSN interaction matrix $(\omega_{ij}, K_{ij})$. The capacity of a coding scheme (Fig. 7a, middle) can be evaluated by its ability to separate odors with similar odor space (Fig. 7a, top) in the corresponding OSN response space (Fig. 7a, bottom). Here, we try to address whether and how the inclusion of inhibitory OSN responses in addition to the excitatory ones can enhance the coding capacity.

To answer these questions, we used Eq. (1) with the parameters $\{R_{\max}, \alpha_j, \omega_{ij}, K_{ij}\}$ determined from the experimentally measured responses of $M = 24$ ORs to $N = 110$ odors by Hallem and Carlson[22] (see "Methods" section for details). For a large ensemble of random odor mixtures with the odor number varying from 5 to 60, we computed the coding capacity in terms of the sum of information entropy of individual OSNs, defined as the expectation of logarithm of probability distribution of odor-evoked responses (see "Methods" section for details). In addition, we calculated the effective coding dimension based on the principle component analysis (PCA) of the odor-evoked responses across OSNs.

As shown in Fig. 7b, the inclusion of odor-evoked inhibition in OSNs increases the information entropy of most OSNs and the total entropy of the whole system for all odor number tested (Fig. 7b, inset). Mechanistically, the inclusion of OSN inhibition increases the coding capacity of OSNs by preventing response saturation (Supplementary Fig. 7) and making the responses more uniformly distributed within the response dynamic range (Supplementary Fig. 8). The correlations in the OSN responses were studied by the PCA. A given principle component (PC) can be used for coding if its variance determined by its eigenvalue is larger than a detection (coding) threshold set by the noise level in the system. The number of PCs whose eigenvalue is above the coding threshold (set to be 1 in this study) defines an effective (independent) dimension for coding. As shown in Fig. 7c, the inclusion of OSN inhibition decorrelates odor-evoked responses across OSNs, thus increasing the independent dimensions of odor coding. Therefore, an OSN with both odor-evoked inhibitory and excitatory responses can compute and amplify the difference between similar odor mixtures as illustrated in Fig. 7a.

## Discussion

A central question in olfaction is how odors are recognized by the olfactory system. Research over the past has focused predominantly on odor-evoked activation of OSNs and established that odors are represented by the combinatorial activation of OSNs[27, 32–35]. Here, we demonstrate that odor-evoked inhibition of *Drosophila* OSNs can directly encode odor identity and drive olfactory perception, and that odors are encoded by both odor-evoked inhibition and activation of OSNs. We report four major findings. First, in flies with only one type of functional OR, odor-evoked inhibition can drive olfactory behaviors, just as odor-evoked activation does. Second, a single type of OSN can drive two opposing behaviors and discriminate odor mixtures if inhibition and activation coexist. Third, genetic disruption of odor-evoked inhibition induces a switch of olfactory behaviors. Fourth, odor-evoked inhibition of OSNs increases odor-coding capacity by reducing response saturation and decorrelating odor representation. These findings establish that odor-evoked inhibition of OSNs is a primary odor code and that a bidirectional code with both inhibition and activation in the same OSN is an efficient coding strategy.

Inhibition of the OSNs can result from diverse mechanisms such as odor blockage of transduction channels[36], competitive binding between odors in a mixture[37], or an inhibitory signaling distinct from odor excitation[17]. Here, we focused on the odor-evoked inhibition of the basal activity in the OSNs. Two alternative mechanisms may drive such an odor- and OR-specific inhibition[17]. Odors might bind to OR proteins, leading to the opening of an inhibitory conductance such as potassium channels[11, 26] to counteract the basal activity. Alternatively, odors might bind to ORs and stabilize them in inactive state[21], thus directly reducing the basal activity of ORs.

In the absence of odor stimulation, we observed a basal inward current in the voltage-clamped OSNs, which increased at higher temperatures but was abolished in the absence of OR/ORCO complex. These results suggest that ORs can be constitutively/thermally activated, producing a basal inward current. This basal current is directly inhibited by inhibitory odors, demonstrated by our finding that inhibitory odors reduce the basal membrane conductance in the voltage-clamped OSNs. A direct inhibition of the OR basal activity is further supported by a lack of odor inhibition of the inward currents mediated by $P2X_2$ channels or ChR2 in the same OSN. In addition, we found that inhibitory odors also inhibit the responses to excitatory odors. Therefore, inhibitory odors can inhibit both spontaneous OR activity and excitatory odor-induced OR activity.

At the functional level, we demonstrated that, similar to odor-evoked activation, odor-evoked inhibition of *Drosophila* OSNs directly encodes perceptual signals and drives olfactory behaviors of both attraction and avoidance. The inclusion of odor-evoked inhibition and activation in the same *Drosophila* OSNs increases odor-coding capacity and also allows neural computation at the level of OSNs before odor information is transferred to the downstream olfactory networks.

Efficient coding with limited sensory channels of limited capacity is a general task for sensory systems[38]. In olfaction, animals use a large repertoire of ORs to discriminate odors[18–20, 39]. A common strategy in olfaction is the use of a combinatorial odor coding based on the activation patterns of ORs[27, 32–35], allowing the discrimination of more odors than the number of OR types. Our experiments reveal another strategy for efficient odor coding by compacting two modes of neuronal responses (i.e., inhibition and activation) in the same OR, hence effectively increasing the number of ORs because the two OR modes can drive opposing behaviors. The existence of odor-evoked inhibition in OSNs from insects to mammals[7–15] suggests that this dual odor coding may not be unique to *Drosophila* olfaction. However, this strategy may be particularly important for *Drosophila* because its array of ORs is much smaller than that of vertebrates[18–20, 39].

Our finding that odor-evoked inhibition in the OSNs can drive olfactory behaviors raises a question of how the brain processes odor-evoked inhibition. Odor-evoked inhibitory responses have been observed in *Drosophila* OSNs[21, 22] and projection neurons of the antennal lobe[40], but they have not been observed in Kenyon cells of the mushroom body[41]. One possibility is that the

inhibitory responses may have been converted into excitatory responses via the local neuron-mediated negative feedback[42] or inhibitory projection neuron-mediated inhibition[43–45]. Alternatively, odor-evoked inhibition may be transmitted through other circuits because the higher olfactory centers in *Drosophila* have not yet been fully characterized[46].

Although spontaneous activity is known to be important in early neural development[47, 48], its widespread existence in adults has remained puzzling[3, 9]. Basal activity could desensitize the excitatory responses[22], but this effect is only modest in *Drosophila* OSNs because it occupies <20% of the OSN response range (Supplementary Table 1). A similar amount of basal activity has also been reported in auditory hair cells[2]. Our results show that such a level of basal activity enables odor-evoked bidirectional responses in single OSNs, which greatly increase the odor-coding capacity. Our findings thus highlight the importance of spontaneous basal activity in sensory coding and perception.

## Methods

**Animals**. All flies were raised on standard cornmeal agar medium, under 60% humidity and a 12-h light/12-h dark cycle at 25 °C. The *Or10a-Gal4* (BL9944), *Or42b-Gal4* (BL9971), *Or43a-Gal4* (BL9974), *Or49b-Gal4* (BL9986), *Or56a-Gal4* (BL9988 and 23896), *Or82a-Gal4* (BL23125), *Or85a-Gal4* (BL23133), *Or92a-Gal4* (BL23140), *UAS-Orco* (BL23145), *Orco*[1] (BL23129), *Orco*[2] (BL23130), *UAS-TNT* (BL28837 and 28997), *UAS-TNT* (inactive, BL28844), *UAS-GCaMP6m* (BL42748 and 42750), *UAS-GCaMP6f* (BL42747), and *UAS-H134R-ChR2* (BL28995) flies were from the Bloomington Stock Center. *UAS-mCD8-GFP* was a gift from Dr. Chris Potter at the Johns Hopkins University School of Medicine, USA. *UAS-P2X$_2$* was a gift from Dr. Zuoren Wang at the Institute of Neuroscience, China. The flies have been backcrossed for seven generations to a laboratory $w^{1118}$ strain.

**Patch-clamp recordings**. Antennal slices were prepared as previously described[23]. Briefly, adult flies were immobilized on ice. The isolated third antennal segment was cut into transverse slices. Slices were stabilized and perfused with 95% O$_2$/5% CO$_2$-bubbled *Drosophila* saline (in mM): 158 NaCl, 3 KCl, 4 MgCl$_2$, 1.5 CaCl$_2$, 26 NaHCO$_3$, 1 NaH$_2$PO$_4$, 5 *N*-tri-(hydroxymethyl)-methyl-2-aminoethane-sulfonic acid (TES), 10 D-glucose, 17 sucrose, and 5 trehalose (pH 7.4). The dissection solution was prepared by replacing NaHCO$_3$, NaH$_2$PO$_4$, and TES with 5 mM 4-(2-hydroxyethl)–1-piperazineethanesulfonic acid (HEPES) and 27 mM NaCl (pH 7.4, adjusted with NaOH, bubbled with O$_2$). All chemicals were from Sigma-Aldrich.

Green fluorescent protein-labeled OSNs were visualized on an upright microscope with an IR-LED (>850 nm) and infrared-differential interference contrast optics. Patch-clamp recordings were performed using MultiClamp 700B. Patch electrodes were filled with intracellular saline (in mM: 185 K-gluconate, 5 NaCl, 2 MgCl$_2$, 0.1 CaCl$_2$, 1 EGTA, 10 HEPES; pH 7.4; ~390 mOsm). For perforated patch-clamp recordings, 200 µg/ml amphotericin B was back-filled into the recording pipette. For the I–V relationship, a cocktail of TTX (50 nM) and TEA (10 mM) was used to block voltage-gated channels. For cell-attached recordings, the recording pipettes were filled with the dissection solution. Signals were digitized and recorded with a Digidata 1440A and pClamp 10.2, filtered at 2 kHz and sampled at 5 kHz. Measured voltages were corrected for a liquid junction potential.

**Odor stimulation for patch-clamp recordings**. Rapid solution changes were effected by translating the laminar flow between two solution streams across the recorded OSN with an electronic SF-77B stepper (Warner Instruments). The solution flow was driven by gravity. Odors were freshly dissolved in *Drosophila* saline daily.

**Optogenetic stimulation**. Flies expressing ChR2[49] in *Or85a*-OSNs were raised in complete darkness on standard medium supplemented with 100 µM all-trans retinal. Antennal slices were prepared as usual. The patch-clamp recordings were conducted in a light-proof Faraday cage. A 480-nm LED (Sutter Instrument) coupled to the epifluorescence port of the Slicescope Pro 6000 (Scientifica) was used to activate ChR2.

**Pharmacogenetic stimulation**. Flies expressing P2X$_2$[50] in *Or85a*-OSNs were raised on standard medium under regular conditions. P2X$_2$ was activated by the application of ATP (1 mM, dissolved in *Drosophila* saline) through the odor-deliver system.

**Temperature change**. Temperature was controlled using a Warner CL-100 (Warner Instruments). The heater and cooler were positioned close to the inlet of the recording chamber, and the perfusion solution flowed through the heater/

cooler. A temperature sensor was positioned ~50 µm from the recorded OSNs for measurement of local temperature around the recorded cells.

**Power spectral analysis**. Under voltage-clamped configuration, continuous recordings lasting minutes were made in the targeted OSNs. The average power density spectrum was calculated in 30-s segments with 50% overlap. The difference between the two spectra of the basal current and the inward current induced by a short pulse of ethyl 3-hydroxybutyrate represented the power spectrum of excitatory responses. The power spectrum of the basal current was calculated as the difference between the spectrum of the basal current and the outward current induced by acetophenone. The power spectrum of the excitatory responses to a long step of ethyl 3-hydroxybutyrate was calculated from the two spectra of the basal current and the steady inward current in responding to a long step of ethyl 3-hydroxybutyrate.

**Calcium imaging**. Adult flies of 1–2 days after eclosion were immobilized on ice, and then stabilized on a piece of 3 M tape (0.6 × 0.6 cm) with the wings and dorsal head glued to the tape. To reduce the brain movement, the proboscis and legs were further stabilized with small stripes of tape. The tape with a stabilized fly was transferred to a recording chamber and the tape was used to seal a square opening (0.4 × 0.4 cm) at the bottom of the chamber (Supplementary Fig. 4). The fly was placed in middle of the chamber opening, facing down the chamber. A small window was opened in the tape with sharp blades to expose the fly head. The leak between the tape and the head was sealed with vacuum grease. Adult-like hemolymph (ALH) without Ca$^{2+}$ was then added to the recording chamber. The regular ALH is composed of (mM): 108 NaCl, 5 KCl, 2 CaCl$_2$, 4 MgCl$_2$, 26 NaHCO$_3$, 1 NaH$_2$PO$_4$, 5 HEPES, 5 trehalose, 17 D-glucose, bubbled with 95% O$_2$/5% CO$_2$. The cuticle and sac covering the antennal lobe were removed with sharp forceps. The recording chamber was transferred to the microscope stage and perfused with regular ALH for either two-photon, or confocal, or wide-field fluorescence calcium imaging.

For odor stimulation, air-phase odors were delivered to the antennae, which positioned under the recording chamber. The tube for a background humidified air flow at a rate of 500 ml/min was positioned ~1 cm away from the fly antennae. A filter paper absorbed 100 µl liquid odor was placed inside a glass tube. An solenoid valve-controlled air flow of 50 ml per min passed through the glass tube and was then mixed with the background air flow.

Calcium imaging was performed with an A1 R multi-photon laser-scanning confocal microscope (Nikon) with a ×60, NA 1.0 water-immersion objective (Nikon), high-sensitivity non-descanned detectors, and a Mai Tai DeepSee ultrafast laser (Spectra-Physics). Time-lapse imaging series of GCaMP6m from a single *z* plane of the targeted glomerulus were acquired at ~7 frames per s with a resolution of 512 × 128 pixels. In some cases, imaging was performed under the confocal mode of the same A1 microscope with a sapphire laser of 488 nm (Coherent), or performed under wide-field fluorescence imaging with a slicescope Pro 6000 (Scientifica) with a ×60, NA 1.0 water-immersion objective (Olympus) and a Zyla sCMOS camera (Andor).

The GCaMP6m fluorescence images[51] were analyzed with the software Nikon NIS-Elements. A mean background was subtracted from the targeted glomerulus. We calculated the odor-evoked fluorescence intensity changes as $\Delta F/F$, where $F$ is the maximal fluorescence intensity, $\Delta F$ is the fluorescence change from the baseline.

**Behavioral assays**. Adult male and female flies were collected within 6 h after eclosion. After 24-h starvation in vials with water-absorbed filter strips, ~70 flies were transferred into the sliding chamber of a T-maze (4M Instrument & Tool LLC). The slider was then lowered, enabling the flies to face the opening of the choice tubes connected. The test tube contained a filter paper strip loaded with 10 µl of test liquid odor (diluted in either paraffin oil or water), unless stated otherwise. The control tube contained a strip loaded with 10 µl of odor solvent. The tubes were prepared 30 min before the behavioral test, allowing odor and solvent partition to equilibrate. The positions of the test and control tubes were alternated for each trial. New flies and tubes were used for each trial. The flies were allowed to choose between the tubes for 2 min in complete darkness and then counted. More than eight trials were repeated for each behavioral test. The preference index, PI, was calculated as the difference between the fly numbers in the test and control tubes divided by their sum. PI = 0 indicates an equal distribution of flies between the two tubes; PI = 1 indicates that all flies were attracted to the test tube; PI = −1 indicates that all flies avoided the test tube. All the behavioral experiments were performed at the circadian time of CT5-CT9.

**Generation of *Or10a* knock-out and *Or85a*$^{Gal4}$ knock-in flies**. The flies were generated using Cas9-mediated gene editing methods described before[52, 53].

For *Or10a* mutant, the two following guide RNAs were used:
*Or10a*-sg1 GACATAATGGGCTATTGGCCGGG
*Or10a*-sg2 GGTGGCCACGCCAATGGCCAGG

To generate *Or85a* Gal4-knock-in donor construct, the left homology arm was amplified with primer *Or85a*-5arm-F and *Or85a*-5arm-R, and the right homology arm was amplified with primer *Or85a*-3arm-F and *Or85a*-3arm-R. The backbone

was amplified from vector pBluescript SK(−) with primers pBF and pBR. The backbone and the homology arms were first linked together with Gibson Assembly Kit as pBS-85aLA-85aRA. The 2A-Gal4-loxP-3xP3 RFP-loxP cassette was cut with restriction enzyme *Not* I and *Asc* I. 85aLA-pBS-85aRA linear DNA was cut from pBS-85aLA-85aRA vector, and combined with the cassette with Gibson Assembly Kit, producing the final donor construct pBS-85aLA-2A-Gal4-loxP-3xP3 RFP-loxP-85aRA.

Primers:pBF 5′–TGGCGTAATCATGGTCATAGC–3′
pBR 5′–CTGGCGTAATAGCGAAGAGG–3′
*Or85a*-5arm-F 5′–CCTCTTCGCTATTACGCCAGACGGCTGGTAGATGGA GTTG–3′
*Or85a*-5arm-R 5′–GGCGCGCCATAAGAATGCGGCCGCAAGGACTGGCT CTTGAATGTACT–3′
*Or85a*-3arm-F 5′–GCGGCCGCATTCTTATGGCGCGCCTTCCACAACAGC AACTCCAAG–3′
*Or85a*-3arm-R 5′–GCTATGACCATGATTACGCCATGAGAACCGCACAGA TTTATGG–3′

Below are the two sgRNAs designed to target the region about 30–240 bp downstream of start codon of *Or85a*, and the third sgRNA targeted the site about 1.6 kb downstream of start codon. The sgRNAs' sequences were as follows.
*Or85a*-sg1 GGATCCTTATTTCGATCCCGGG
*Or85a*-sg2 GTTCAAGAACTTCACGACCACGG
*Or85a*-sg3 GCCCGTCTGAAACTGCCGTCCGG

Both the *Or10a* mutant and *Or85a*^*Gal4* knock-in flies were validated by sequencing.

**Computation modeling and analysis.** Suppose that there are $N$ odors in a mixture, with the concentration of odor $i$ (=1, 2, 3..., $N$) given by $C_i$. The odor-evoked responses in OSNs are determined by the odor–OR interactions[21, 22]. Because the olfactory transduction in *Drosophila* OSNs remains controversial[54], we used a simple two-state model to describe the odor-evoked response in OSNs (Supplementary Fig. 9). In this model, ORs have two states: an inactive state that leads to no OSN activity and an active state that leads to a maximum activity $R_{max}$ in the OSNs. Odor molecules bind to ORs and modulate the transition rates between these two OR states, with excitatory odors stabilizing the active states and inhibitory odors stabilizing the inactive states. We set two binding constants for each odor-OR/OSN pair $(i, j)$: $K_{I,ij}$ is the dissociation constant for the inactive OR state and $K_{A,ij}$ is the dissociation constant for the active state. We modeled the responses of OSNs to odor mixtures by using two modes of odor–OR/OSN interactions: multiple binding sites, in which different odors bind to different sites, and competitive binding, in which different odors compete for the same binding sites. We show below that these two cases produce similar results.

Multiple binding sites: We constructed the model based on transition kinetics between the two functional states (active and inactive) and the different odor binding states (bound and unbound). For steady-state properties, the effective "free energy" difference between the inactive and active states of the OR/OSN$_j$, $\Delta F_j$, depends on the odor concentrations:

$$\Delta F_j \equiv F_{j,I} - F_{j,A} = E_{0,j} + \sum_i \left[ \ln\left(1 + \frac{C_i}{K_{A,ij}}\right) - \ln\left(1 + \frac{C_i}{K_{I,ij}}\right) \right] \quad (2)$$

where $E_{0,j}$ is the free energy difference in the absence of any odors, and the other terms correspond to the entropic contributions because OR can be either vacant or bound by an odor molecule. The average activity of the OR/OSN$_j$, $R_j$, can then be written as

$$R_j = \frac{R_{max}}{1 + \exp(-\Delta F_j)} = R_{max}\left[1 + \alpha_j \prod_i \frac{1 + C_i/K_{I,ij}}{1 + C_i/K_{A,ij}}\right]^{-1} \quad (3)$$

where $\alpha_j = \exp(-E_{0,j})$, and the baseline activity in the absence of any stimulus is $R_0 = R_{max}/(1 + \alpha_j)$. This energetic approach gives the same results as solving the steady state of the kinetic equations, as shown below.

For excitatory odors, we have $K_{A,ij} < K_{I,ij}$, which indicates that they bind to the active OR state with a higher affinity. Inhibitory odors stabilize the inactive OR state, that is, $K_{I,ij} < K_{A,ij}$. For simplicity, we assume the excitatory odor only binds to the active OR state and the inhibitory odor to the inactive OR state. That is, $K_{I,ij} = \infty$ for excitatory odors, and $K_{A,ij} = \infty$ for inhibitory odors. Then, we have

$$R_j = R_{max}\left[1 + \alpha_j \prod_{i=1}^N \left(1 + \frac{C_i}{K_{ij}}\right)^{-w_{ij}}\right]^{-1} \quad (4)$$

where $w_{ij} = \{1, -1, 0\}$ for excitatory, inhibitory, and null responses, respectively; $K_{ij}$ represents the finite dissociation constant $K_{I,ij}$ for inhibitory odors and $K_{A,ij}$ for excitatory odors.

Competitive binding: The free energy difference between the inactive and the active states of the OR/OSN$_j$ is

$$\Delta F_j \equiv F_{j,I} - F_{j,A} = E_{0,j} + \ln\left(1 + \sum_q \frac{C_q}{K_{qj}}\right) - \ln\left(1 + \sum_p \frac{C_p}{K_{pj}}\right) \quad (5)$$

where $p$ and $q$ represent the inhibitory and excitatory odors, respectively. The activity of OR/OSN$_j$ corresponding to Eq. 4 is

$$R_j = R_{max}\left[1 + \alpha_j \frac{1 + \sum_{p=1}^{n_i} C_p/K_{pj}}{1 + \sum_{q=1}^{n_e} C_q/K_{qj}}\right]^{-1} \quad (6)$$

where $n_e$ and $n_i$ are the numbers of excitatory and inhibitory odors, respectively.

Most of the results presented in the main text were obtained using the model of multiple binding sites. However, the model with competitive binding as described in Eq. 6 generates qualitatively similar results (see below).

Advantages of encoding with bidirectional odor-evoked responses: Odors in the natural environment vary in both their frequencies of appearance and their concentrations. In addition, some odors may appear together (or correlated). An optimal olfactory system should be able to make two chemical mixtures or two vectors in the odor space ($C_1$ and $C_2$) distinguishable in the OSN response space, that is, make the OSN activity vectors ($R_1$ and $R_2$) separable. Essentially, for the distribution of points in the odor space $P(C)$, the coding transforms it to the distribution of OSN activity $P_n(\vec{R})$. When $P_n(\vec{R})$ is uniform, it encodes the maximum amount of information.

We compared the coding capacity (in terms of entropy) and de-correlation (in terms of the principal component spectrum) of OSN responses for cases with and without odor-evoked inhibitory responses. The distribution of odor mixtures in the fly's natural environment is unknown. Here, we used the ensemble of odors that have been comprehensively studied in the literature by the Carlson lab[22] to compose the odor mixtures.

Setting the parameters based on experimental data: All the parameters {$R_{max}$, $\alpha$, $\omega_{ij}$, $K_{ij}$} are determined according to the measured responses of 24 ORs to 110 odors[22], where responses were coarse-grained (digitalized) to 6 levels: $\Delta R = \{-1, 0, 1, 2, 3, 4\}$, in which −1 denotes inhibition and positive numbers represent various degrees of excitation. We ignore the three odors that elicited no OR/OSN responses, and the three ORs/OSNs that showed only inhibitory or no responses. To calculate the OR/OSN responses to odor mixtures according to Eq. 4 or Eq. 6, we defined the discrete response, $\Delta R$, based on the spiking rates $R_{spike}$ according to the criteria used by others[22].

$$\Delta R = \begin{cases} -1, & R_{spike} \le 15 \\ 0, & 15 < R_{spike} < 50 \\ 1, & 50 \le R_{spike} < 100 \\ 2, & 100 \le R_{spike} < 150 \\ 3, & 150 \le R_{spike} < 200 \\ 4, & R_{spike} \ge 200 \end{cases} \quad (7)$$

For simplicity, we set the baseline firing rate of all the ORs/OSNs as $R_0 = 30$ spikes per s (and the maximum firing rate of 250 spikes per s), which leads to a constant $\alpha = \frac{R_{max}}{R_0} - 1 = \frac{22}{3}$. The concentration was set as 1 in an arbitrary unit. For excitatory odors, we set the $\frac{1}{K_{ij}} + 1 = \frac{1}{4}\alpha, \frac{2}{3}\alpha, \frac{3}{2}\alpha, 4\alpha$, corresponding to the response states 1, 2, 3, and 4, respectively. For inhibitory odors, we selected the inhibition strength such that the average effect from excitatory odors is roughly balanced by that from inhibitory odors. For simplicity, we assume $K_{ij} = K_j$; that is, the inhibitory strength is the same for any inhibitory odor–OSN$_j$ pair, where $N_{0j}$ is the number of inhibitory odors for OSN$_j$. The value of $K_j$ is then set by requiring a rough balance of inhibitory and excitatory stimuli on average:

$$N_{0j} \ln\left(1 + \frac{1}{K_j}\right) - \sum_{q=1}^{N_{1j}} \ln\left(1 + \frac{1}{K_{qj}}\right) = -\ln\left(\beta_j\right) \quad (8)$$

where $N_{1j}$ is the number of excitatory odors of the OSN$_j$ and $\beta_j$ is a constant. For a larger $\beta_j$, the average response becomes larger. In the following simulations, we set it as 1. Other values were also used without changing the general results.

The coding capacity for odor mixtures: We used two methods to compose the odor mixture: the sampling method and the enumeration method. With the sampling method, we fixed the number of odors in the mixtures and randomly sample from the 107 odors 100,000 times. The concentration of each chosen odor was set as 1, and other concentrations were also used without changing the results qualitatively. For each odor mixture, we calculated the responses of the 21 ORs/OSNs. The Shannon entropy of each OR/OSN is computed by $H_j = -\int_{R_{min}}^{R_{max}} P_j(R)\left[\log_2 P_j(R)\right]$, where $P_j(R)$ is the distribution function of the response $R_j$ for the OSN$_j$. In our calculation, this probability is approximated by dividing the whole response range into bins and counting the numbers of the responses that fall into each bin of 1 spike per s. The integration was substituted by summation. The maximum entropy per OR/OSN is $H_{max} = \log_2 (R_{max} - R_{min}) \approx 8$ bit.

For the enumeration method, we assume that any given odor has a probability $p$ to appear in the mixture. For example, $p = 0.2$ leads to roughly $N_{ave} \sim 21$ odors in the mixture. For a given neuron, we denote $N_0$ as the total number of inhibitory odors with dissociation constant $K_0$ and $N_l$ as the total number of "type-$l$" excitatory odors with dissociation constant $K_l$ for $l = 1, 2, 3, 4$ in the odor repertoire. Only a subset of these odors are present in a given mixture. For a

mixture with $N_0$ inhibitory odors and $N_l$ "type-$l$" excitatory odors, the response activity is

$$R\left(\bar{n}\right) = R_{\max}\left[1 + \alpha\frac{(1 + C_0/K_0)^{n_0}}{\prod_{l=1}^4 (1 + C_0/K_l)^{n_l}}\right]^{-1} \tag{9}$$

where $\bar{n} = (n_0, n_1, n_2, n_3, n_4)$ characterizes the odor mixture. The probability of this random mixture characterized by $\bar{n}$ is given by

$$P\left(\bar{n}\right) = \prod_{l=0}^4 \binom{N_l}{n_l} p^{n_l}(1 - p)^{N_l - n_l} \tag{10}$$

where the range of $n_l$ is from 0 to $N_l$. For each choice of $\bar{n}$, we computed the corresponding $R_{\bar{n}}$ and $P\left(\bar{n}\right)$, from which we get the distribution $P(R)$ for this neuron exactly without sampling. The summation of all the entropy was computed for cases with or without odor-evoked inhibitory responses. The entropy of each OR/OSN and the total entropy for all ORs/OSNs is much larger when including odor-evoked inhibitory responses (Supplementary Figs. 7 and 8). Including inhibitory response also reduced the average response, thus avoiding OR/OSN saturation (Supplementary Fig. 7) and making the OR/OSN responses more uniform (Supplementary Fig. 8). These two effects both increase the coding capacity of ORs/OSNs.

Results from the sampling method (Supplementary Fig. 10a, b) were consistent with the enumeration method (Supplementary Fig. 10c, d). We also computed the total entropy of all the ORs/OSNs for the competitive binding case (Supplementary Fig. 10e–g), which gave results similar to those in Fig. 7b.

Principal component analysis: For the 100,000 randomly sampled odor mixtures, we first transformed the response of OR/OSN into discrete states according to Eq. 7, producing 100,000 points in a 21-dimension space. We then performed the principal component analysis by constructing the correlation matrix of the 21 ORs/OSNs. The eigenvalue of a given PC characterizes the variation of OSN responses along that PC direction. The higher the eigenvalue, the more information can be coded in that PC. A PC can be used effectively for coding when its eigenvalue is larger than a threshold determined by noise. The number of PCs with eigenvalues above the noise threshold is defined as the effective coding dimensions. A higher effective coding dimensions corresponds to more independent directions for odor representation by ORs/OSNs. The noise threshold is set to be 1 in our study. The PCA results using competitive binding (Supplementary Fig. 10f) were similar to those using multiple binding sites (Fig. 7c).

Equivalence of the kinetic approach and energetic approach: Both the kinetic and the energetic approaches can be used to describe the odor responses in OSNs. We used the energetic approach in the main text because it is easier to generalize to odor mixtures. The kinetic approach (Supplementary Fig. 9) can yield additional time-dependent information. However, this information is not considered in this study. In the following section, we determine the steady-state response by solving the kinetic equations. We first consider the simple situation with only one odor. As shown in Supplementary Fig. 9, an OR/OSN$_j$ can exist in two functional states: the active and the inactive states labeled by $R_j^*$ and $R_j$, respectively. In addition, a receptor can be either bound or unbound by odor molecules $L_i$. Therefore, there are four microscopic states: $R_j, L_i \cdot R_j, L_i \cdot R_j^*, R_j^*$, denoted as 1, 2, 3, and 4, respectively, for simplicity. The ligand binding/unbinding kinetics and the active/inactive kinetics among the four states are illustrated in Supplementary Fig. 9 with their rates specified. The kinetics of the four microscopic states can be described by the following rate equations:

$$\frac{dP_1}{dt} = k'_{\text{off}}P_2 - k'_{\text{off}}\frac{C_i}{K_{I,ij}}P_1 - \omega P_1 + \omega\alpha P_4 \tag{11}$$

$$\frac{dP_2}{dt} = k'_{\text{off}}\frac{C_i}{K_{I,ij}}P_1 - k'_{\text{off}}P_2 - \omega'P_2 + \omega'\alpha'P_3 \tag{12}$$

$$\frac{dP_3}{dt} = k_{\text{off}}\frac{C_i}{K_{A,ij}}P_4 - k_{\text{off}}P_3 - \omega'\alpha'P_3 + \omega'P_2 \tag{13}$$

$$\frac{dP_4}{dt} = k_{\text{off}}P_3 - k_{\text{off}}\frac{C_i}{K_{A,ij}}P_4 - \omega\alpha P_4 + \omega P_1 \tag{14}$$

where $p_n(t)$ is the probability in a given state $n$ (=1, 2, 3, 4) at time $t$ with the sum $P_1 + P_2 + P_3 + P_4 = 1$.

Under steady-state conditions, that is, $dP_n/dt = 0$ ($n = 1, 2, 3, 4$), and with the detailed balance condition $\alpha'/K_{A,ij} = \alpha/K_{I,ij}$ satisfied, the steady-state probabilities of the four microscopic states can be obtained by requiring that the forward and backward fluxes be equal for each transition pair:

$$k'_{\text{off}}P_2 = k'_{\text{off}}\frac{C_i}{K_{I,ij}}P_1 \tag{15}$$

$$\omega'P_2 = \omega'\alpha'P_3 \tag{16}$$

$$k_{\text{off}}\frac{C_i}{K_{A,ij}}P_4 = k_{\text{off}}P_3 \tag{17}$$

$$\omega\alpha P_4 = \omega P_1 \tag{18}$$

By solving the above equations, we obtain $P_1, P_2, P_3, P_4$. The total probability of receptors being in the active state (both ligand bound and unbound) is

$$P_{\text{active}} = P_3 + P_4 = \frac{1 + \frac{C_i}{K_{A,ij}}}{\alpha\left(1 + \frac{C_i}{K_{I,ij}}\right) + \left(1 + \frac{C_i}{K_{A,ij}}\right)} = \left[1 + \alpha\frac{1 + \frac{C_i}{K_{I,ij}}}{1 + \frac{C_i}{K_{A,ij}}}\right]^{-1} \tag{19}$$

which makes the average activity of the OR/OSN

$$R_j = R_{\max}P_{\text{active}} = R_{\max}\left[1 + \alpha\frac{1 + \frac{C_i}{K_{I,ij}}}{1 + \frac{C_i}{K_{A,ij}}}\right]^{-1} \tag{20}$$

The above result is the same as Eq. 3 obtained from the effective free energy difference given by Eq. 2. To generalize to the case with multiple odors acting on the same OR/OSN, the energetic approach is much easier to use because the free energy difference is additive, see Eq. 2. Of course, the same result can be obtained from solving the steady state of the kinetic equations. Here, we briefly show the case for two odors (indexed as $i, k$), where there are four different active states due to the bound/unbound states of the two ligands. The total probability of being active is then

$$P_{\text{active}} = \frac{1}{\alpha}\left(1 + \frac{C_i}{K_{A,ij}} + \frac{C_k}{K_{A,kj}} + \frac{C_i}{K_{A,ij}}\frac{C_k}{K_{A,kj}}\right)P_1$$

$$= \frac{\frac{1}{\alpha}\left(1 + \frac{C_i}{K_{A,ij}}\right)\left(1 + \frac{C_k}{K_{A,kj}}\right)}{\frac{1}{\alpha}\left(1 + \frac{C_i}{K_{A,ij}}\right)\left(1 + \frac{C_k}{K_{A,kj}}\right) + \left(1 + \frac{C_i}{K_{I,ij}}\right)\left(1 + \frac{C_k}{K_{I,kj}}\right)} \tag{21}$$

$$= \left[1 + \alpha\frac{(1 + C_i/K_{I,ij})(1 + C_k/K_{I,kj})}{(1 + C_i/K_{A,ij})(1 + C_k/K_{A,kj})}\right]^{-1}$$

where $P_1$ is the probability of the inactive state without any odor binding. The above equation can be generalized to $N$ odors, with $2^N$ different odor (ligand) binding combinations for the active state of the receptor. The corresponding active probability is

$$P_{\text{active}} = \frac{1}{\alpha}\prod_{i=1}^N\left(1 + \frac{C_i}{K_{A,ij}}\right)P_1 = \frac{\frac{1}{\alpha}\prod_{i=1}^N\left(1 + \frac{C_i}{K_{A,ij}}\right)}{\frac{1}{\alpha}\prod_{i=1}^N\left(1 + \frac{C_i}{K_{A,ij}}\right) + \prod_{i=1}^N\left(1 + \frac{C_i}{K_{I,ij}}\right)} \tag{22}$$

$$= \left[1 + \alpha\prod_{i=1}^N\frac{(1 + C_i/K_{I,ij})}{(1 + C_i/K_{A,ij})}\right]^{-1}$$

and the average activity of the OSN$_j$ is

$$R_j = R_{\max}\left[1 + \alpha\prod_{i=1}^N\frac{(1 + C_i/K_{I,ij})}{(1 + C_i/K_{A,ij})}\right]^{-1} \tag{23}$$

which is exactly Eq. 3.

For the competitive binding interaction, that is, all ligands compete for the same binding sites, it is easy to follow the same analysis as above. Given $N$ odors, there are $2(N + 1)$ microstates, and half of them correspond to active state. The corresponding active probability is

$$P_{\text{active}} = \frac{1}{\alpha}\left(1 + \sum_{i=1}^N\frac{C_i}{K_{A,ij}}\right)P_1 = \frac{\frac{1}{\alpha}\left(1 + \sum_{i=1}^N C_i/K_{A,ij}\right)}{\frac{1}{\alpha}\left(1 + \sum_{i=1}^N C_i/K_{A,ij}\right) + \left(1 + \sum_{i=1}^N C_i/K_{I,ij}\right)} \tag{24}$$

$$= \left[1 + \alpha\frac{1 + \sum_{i=1}^N C_i/K_{I,ij}}{1 + \sum_{i=1}^N C_i/K_{A,ij}}\right]^{-1}$$

and the average activity of the $OSN_j$ is

$$R_j = R_{max} P_{active} = R_{max} \left[ 1 + \alpha \frac{1 + \sum_{i=1}^N C_i/K_{I,ij}}{1 + \sum_{i=1}^N C_i/K_{A,ij}} \right]^{-1} \qquad (25)$$

which is exactly Eq. 6. Therefore, the energetic approach and the kinetic approach lead to the same results.

**Statistics**. All experiments were performed with experimental and control groups in parallel. Sample size was determined based upon preliminary experiments. Data were analyzed statistically using one-way ANOVA $t$ test, presented as mean ± SEM, unless otherwise stated.

**Data availability**. All relevant data supporting the findings of this study are available from the corresponding author on request.

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

## Acknowledgements

We thank K.-W. Yau, J. Carlson, M. Luo, Z. Wang, C.J. Potter, W.W. Yue, Y. Lin, M.-H. Han, L. Lai, M. Rigotti, H.-X. Ren and Y. Naya for discussions or comments on the

manuscript, and A. Kolodkin, H. Jiang and V. Bhandawat for technical help. The work was supported by National Natural Science Foundation of China (31471053, 31671085, and 91430217), the Ministry of Education (the Young Thousand Talent Program (D.-G.L.)), Ministry of Science and Technology (2015CB910300), and the State Key Laboratory of Membrane Biology. Y.T. was supported by NIH Grant R01GM081747. Opinions, interpretations, conclusions, and recommendations are those of the author and are not necessarily endorsed by the US Army.

## Author contributions

L.-H.C. and D.-G.L. conceived and designed the study. L.-H.C., D.Y., B.-Y.J. and D.-G.L. performed electrophysiological recordings. X.Z. and D.-G.L. performed fly genetics. W.W. performed behavioral experiments. B.-Y.J. and M.-T.L. performed calcium imaging. Y.T., S.Q., C.T. and D.-G.L. performed computation analyses. L.-H.C., Y.T., X.Z. and D.-G.L. wrote the manuscript.

## Additional information

**Competing interests:** The authors declare no competing financial interests.

