## [Peer Review File · Nature Communications]

Reviewers' expertise:

Reviewer #1: Information coding in the olfactory system, computational modeling;

Reviewer #2: Drosophila olfactory circuits;

Reviewer #3: Olfactory coding in Drosophila.

Reviewers' comments:

Reviewer #1 (Remarks to the Author):

The authors examine the role of inhibitory responses from peripheral OSNs in olfactory coding in Drosophila. Using a combination of genetic and electrophysiological tools, the authors determine that (1) odor-evoked inhibitory responses are formed by blocking spontaneously open channels; (2) odor-evoked inhibition can determine behavior in both mutants expressing one functional receptor and wild type animals expressing an array of receptors; (3) single OSNs can also drive dichotomous behavior based on excitatory or inhibitory odorants. Further, the authors use computational modeling to assess how inhibition may increase odor-coding space.

The study investigates an interesting topic with a well-designed biological approach. The role of odor-evoked inhibition in OSNs is an exciting new topic in odor coding. The authors present a logical flow of ideas asserting the role of odor-evoked inhibitory responses in governing behavior. The text is well-written and the majority of the assertions are well-supported. What I found to be rushed and incomplete is a discussion of how inhibition contributes to odor discriminability using computational modeling. Eight pages of methods describing the computational model is represented in the main text by a single paragraph which only asserts and does not justify their conclusions. For example, "the inclusion of OSN inhibition decorrelates odor-evoked across OSNs (Fig. 7c)" is the entire explanation of decorrelation. Further, I find figure 7 difficult to interpret without more information.

I would suggest one of two paths. One option, would be to significantly expand the text and figures regarding the modeling work so that the conclusions are well supported by interpretable results; or that a much smaller simpler model with simplified conclusions be presented here and the more complex model with complex conclusion be saved for a second short paper.

SPECIFIC COMMENTS

- 53: exciting
- 61: olfaction in Drosophila
- Figure 1b: Label individual traces of odor stimulation
- Figure 2a: E-3 in 3. Pulse is hard to see. It might be worth mentioning in the Figure or caption.
- Figure 2d: Use distinguishable dots for excitatory and inhibitory V-I curves.
- Figure 2f: concentrations should be shown in figure or caption.
- Figure 3a (bottom): use lines to connect box in fig. 3a middle to fig. 3a bottom.
- Figure 7: The results given are modeling or experiment?? This is unclear in the caption.
- Figure S9: You should distinguish excitatory odorants (Ci) from inhibitory odorants, perhaps Ci'.

Reviewer #2 (Remarks to the Author):

This paper uses patch clamp recordings from Drosophila OSNs and behavioral assays to examine the mechanisms underlying odor-evoked inhibition of OSNs and the functional consequences of this inhibition for olfactory behavior. Although the idea that inhibitory odorants close or stabilize

spontaneous openings of olfactory receptors has been prevalent in the literature, this represents the first direct demonstration of this fact to my knowledge. Overall I think this paper is interesting, well-written, and carefully done. I have a few questions/comments I think the authors should address before the paper is accepted.

1) The Methods (and the previously cited paper) indicate that the recordings were done with OSNs submerged in water, however, this is not indicated in the main text of the results. Were the cell-attached recordings also done in water or in air? To what degree was the sensillum lymph intact versus disrupted and what effect does dialyzing this compartment have on spontaneous activity of OSNs?

2) I was not sure of the significance of the power spectra shown in Figures 2a-c. Since these are whole-cell recordings, not single channels, it is not clear to me that one would expect the noise characteristics of the macroscopic signal to be different even if the channels underlying excitation and inhibition were different. It seems that this analysis mostly quantifies recording noise.

3) The demonstration of competitive antagonism in figure 3e is very nice and a strength of the paper. Does this finding generalize to other inhibitory odors? Or are there different modes of action for the effects of different inhibitory odorants?

4) Overall the behavioral experiments in figures 4 and 5 appear to be consistent with a model in which different olfactory glomeruli contribute towards attraction or aversion with different weights (e.g. Badel and Kazama 2016). So, for example if activity of Or56a has a negative valence, then activation of this glomerulus by geosmin increases avoidance while inhibition of this glomerulus by linalool increases attraction. The fact that reduction of activity in an aversive glomerulus leads to attraction could be a result of the two-choice assay. I wonder if it would clarify the results to speak about the valence of a glomerulus (which can be either activated or suppressed by different odorants) rather than talking about the actions of specific inhibitory or excitatory odors on behavior. This seems to be a simpler explanation of the behavior than conversion of inhibition into excitation by feedback, as suggested in the discussion.

5) The presentation of the model is very brief and therefore opaque. Is the essential insight of the model that the possibility of inhibitory responses decreases saturation, or simply that it increase the dynamic range of each OR channel (because the signal can go from 0 to R_{max} , instead of R_0 to R_{max}). Is it necessary to model bindings and transitions, as described in the methods, to reach these conclusions? I think it would be helpful to reader if the authors either presented a fuller picture of the full model in the results, including what assumptions it makes, or simplified the model so it captures only the most important intuitions. The full model would potentially be better suited for a separate paper in another journal.

Minor comments:

Figure 2e is difficult to see at its current scale. Figure 3a might be easier to parse if the different panels were at similar scale.

Figure 1a looks as though the acetophenone response is saturating at $\sim 6-8$ pA, which is much less than the average basal current of -18 pA. Perhaps it would be useful to report or show the distribution of basal current amplitudes?

Reviewer #3 (Remarks to the Author):

In "Odor-evoked inhibition of olfactory sensory neurons drives olfactory perception in *Drosophila*", authors Cao et al address an interesting and fundamental question: do odor-evoked pauses in spontaneous spiking in olfactory sensory neurons carry information about the odor? This ambitious study uses a variety of genetic manipulations, recording techniques, behavioral tests, and modeling to show that odor-evoked inhibition, like excitation, can carry information about odors. The observation of odor-evoked inhibition in OSNs is not new, so it is perhaps not surprising that these responses carry useful information. But the thorough, multi-level analysis reported here is impressive and should interest a wide range of readers interested in sensory coding.

Major concerns:

(1) As detailed below, most results are reported as single examples, and no results are supported by necessary descriptions of variance and statistical significance. The authors need to check each figure panel and provide appropriate statistical support for each claim they make.

(2) The authors used a new procedure to deliver odor stimuli in the aqueous phase while recording from OSNs in dissected antennae. As detailed below, their results show levels of spontaneous activity substantially different from those reported by other groups recording from the same neurons. This disparity is not mentioned by the authors. The potential concern here is that the authors' new preparation may not be functioning in a normal, physiological fashion. This concern is amplified by comparisons the authors make between results obtained through their new procedure in isolated antennal preparations and behavioral results obtained from intact animals. The authors need to directly address this concern: does their new preparation function in a normal, physiological fashion, and, if its properties are somehow different from those of intact animals, how can the authors compare results obtained this way to results obtained from intact animals?

Minor concerns:

(1) The writing is generally very clear, but the manuscript would benefit from light editing to correct small grammatical errors and ambiguities.

(2) title and elsewhere: "Inhibition": Throughout the text, the authors argue that inhibition carries information, just like excitation. However, in most places, the inhibition that matters here is inhibition in a narrower meaning: reduction of basal firing rate. The authors might want to distinguish this kind of inhibition from the hyperpolarization that can be observed in intracellular recordings.

(3) line 46-7 and elsewhere: "a dual code": This phrasing is misleading because it suggests the olfactory system is using two separate mechanisms to encode information rather than a single mechanism that employs bidirectional changes in firing rate.

(4) line 58: "independent sensory code": again, the authors provide no evidence that inhibition and excitation are evaluated downstream by independent mechanisms.

(5) line 74-5: "Notably, a loss of odor-evoked inhibition can result in a complete switch of olfactory behaviors." This sentence does not accurately describe the experimental result. In the behavioral experiments, the authors abolished not only odor-evoked inhibition but also spontaneous activity (transmission, more accurately). It's not established whether abolishing spontaneous activity has its own effect on coding. (It might, for example, if the lack of spontaneous firing effectively raises the firing threshold of follower cells.)

(6) line 91-2: "Correspondingly, an outward receptor current was induced by acetophenone...." Later the authors show the outward current is actually a reduction of inward currents. The wording used here may confuse some readers.

(7) Figure 1a, bottom: What is the rationale for holding the membrane potential at -80mV? Does the spontaneous spike rate measured in whole cell mode match the spike rate recorded with sensillum recordings?

(8) Figure 1b: Is this figure from a single representative cell? The authors need to establish the reliability of their results by showing statistical support for their claims.

(9) line 108: typo: "Supplementary Fig. 1c,e" should be "Fig. 1c and FigS1e"

(10) lines 116-118: "Alternatively, acetophenone may interact with Or85a receptors to close ion channels that are constitutively open and have a reversal potential above the resting potential of OSNs." But this possibility has already been excluded by experiments with the Orco mutant, which are described above. Further, do the authors mean to argue that the ion channels by themselves are constitutively active or that the constitutive activities of ion channels are conferred by the constitutive activity of Or85a? The experiments described below do not clearly differentiate these two possibilities.

(11) line 138: Fig. 2b,c: the authors should show examples from multiple cells.

(12) line 145: "mirrored": this word is not appropriate; it means that, if one of the curves were reflected along the x-axis then the two curves would be superimposable, and they are not. How about something like: "exhibited similar, although opposite in direction, current-voltage relationships".

(13) line 146: Fig. 2d: the authors should show examples from multiple cells. Also, data points shown on the I-V curve should be color-coded to indicate whether they were evoked by excitatory or inhibitory odorants.

(14) line 154: Fig. 2e: Although this effect looks consistent across cells the authors should quantify the variability with standard statistics: show a p-value.

(15) line 164: Figs. 3a: These measurements need to be quantified across cells with appropriate statistics.

(16) line 166: Fig. 3b: These measurements need to be quantified across cells with appropriate statistics.

(17) line 168: Fig. 3c: These measurements need to be quantified across cells with appropriate statistics.

(18) line 169: Fig. 3d: These measurements need to be quantified across cells with appropriate statistics.

(19) line 175: "significantly": the authors should not use this word unless it is accompanied by statistical validation, which is not provided here.

(20) line 176: Fig. 3e: The authors show some numbers in Table 1, but the appropriate statistics must be calculated and shown in the text or figure caption.

(21) line 191: "cell-attached recordings": It would be preferable to replace the cell-attached results with those from sensillum recordings, a technique allowing for more naturalistic experimental conditions. This is especially important because the spontaneous spike rates reported here do not match published sensillum recordings from John Carlson's group; (Spontaneous spike rate in sensillum recordings (Hallem et al 2006, Kreher et al, 2008): Or82a: 16Hz; Or10a: 14Hz; Or42b: 7Hz; Or43a: 21Hz; Or85a: 14Hz). The authors need to address these disparities in spontaneous activity rates.

(22) line 246ff: "Similar results were obtained by disrupting the signaling of odor-evoked inhibition in Or10a-OSNs." But for all results shown in Fig.6, transmission or activity of ORNs were silenced. As noted above, it is unclear whether the observed behavioral effects were caused by abolishing the odor-evoked inhibition of basal activity, or by abolishing basal activity itself. It is unclear how the basal spiking of ORNs affects the fly's ability to follow odor presentations. The authors should discuss this caveat.

(23) line 291: "Third, genetic disruption of odor-evoked inhibition...." Strictly speaking, synaptic transmission was disrupted, including spontaneous transmission, not only inhibition.

(24) line 328-9: "hence effectively doubling the number of ORs because the two OR modes can drive opposing behaviors." Because it has not been established that inhibition of baseline activity provides the same information capacity as excitatory responses, "doubling the number of ORs" is not an accurate comparison.

(25) lines 341-2: "Although spontaneous activity is known to be important in early neural development, its widespread existence in adults has remained puzzling." This issue was investigated by Joseph et al 2012; these authors suggest spontaneous activity is an inevitable consequence of the extreme sensitivity of the OSNs to odors, and that any modification to reduce the spontaneous activity might also reduce sensitivity.

Point-by-point response:

Reviewer #1:

We are grateful to Reviewer 1 for describing the paper as “well-designed”, “an exciting new topic in odor coding”, and “well written”.

We followed the reviewer’s suggestion and expanded the text regarding the modeling work to make it clearer to the reader (see Text in RED at Page 13-15).

Specific comments:

1) 53: exciting

Response: Thanks. Corrected.

2) 61: olfaction in *Drosophila*

Response: Corrected.

3) Figure 1b: label individual traces of odor stimulation

Response: Odor stimulation is given in the legend.

4) Figure 2a: E-3 in 3. Pulse is hard to see. It might be worth mentioning in the Figure or caption.

Response: Following your suggestion, we mentioned it in the caption.

5) Figure 2d: use distinguishable dots for excitatory and inhibitory V-I curves.

Response: Done.

6) Figure 2f: concentration should be shown in figure or caption.

Response: Thanks. We rearranged this panel.

7) Figure 3a (bottom): use lines to connect box in fig.3a middle to fig.3a bottom.

Response: Done.

8) Figure 7: The results given are modeling or experiment? This is unclear in the caption.

Response: Done. We made it clear in the caption that these results are modeling based on the published data by Carlson’s group.

9) Figure s9: you should distinguish excitatory odorant (C_i) from inhibitory odorants, perhaps C_i' .

Response: In this model, an odorant, C_i , can bind to both the inactive and active OR states with constants of $K_{I,ij}$ and $K_{A,ij}$, respectively. Whether an odorant is excitatory or inhibitory is determined by the relative values of $K_{I,ij}$ and $K_{A,ij}$. When $K_{I,ij} \gg K_{A,ij}$, C_i is inhibitory.

Reviewer #2:

We are delighted that Reviewer 2 considers our manuscript as “interesting”, “well-written”, “carefully done”, and “the first direct demonstration”.

Major comments:

1) Were the cell-attached recordings also done in water or in air? To what degree was the sensillum lymph intact versus disrupted and what effect does dialyzing this

compartment have on spontaneous activity of OSNs?

Response: The cell-attached recordings were done in water. In our recordings, we selected the “intact” OSNs, whose dendrites projected into the sensillum (based on their GFP expression). The sensillar hair that housed the OSN dendrite was also intact. These structure features helped maintain the sensillum lymph as intact as possible. However, whether the sensillum lymph would be dialyzed through the odor-entrance pore tubules on the sensillar hair is unclear. Nonetheless, we observed similar spontaneous activity (see Fig. R1) in both air-phase single-sensillum recordings (SSRs) and water-phase cell-attached recordings on the Or85a-OSNs of WT flies.

Fig. R1. (A) Single-sensillum recordings on a live fly. The ab2 sensillum is identified based on the GFP expression driven by Or85a-Gal4. There are two types of spikes of different amplitudes as shown at the bottom. The large (A) and small (B) spikes are from Or59b- and Or85a-OSN, respectively (Hallem EA, et al., 2004, Cell). (B) The counting of the B-type spike responses. The firing rate is 5 Hz (150/30 sec). The average spontaneous firing rate is 5.4 ± 0.9 Hz ($n = 7$; mean \pm S.E.M.). (C) Cell-attached recordings on an Or85a-OSN from an antennal slice. The cell is identified based on its GFP expression driven by Or85a-Gal4. The firing rate is 5.2 Hz (157/30 sec). The average spontaneous firing rate is 4.5 ± 0.9 Hz ($n = 6$). There is no significant difference between SSR and cell-attached recordings ($P = 0.43$).

2) I was not sure of the significance of the power spectra shown in Figures 2a-c. Since these are whole-cell recordings, not single channels, it is not clear to me that one would expect the noise characteristics of the macroscopic signal to be different even if the channels underlying excitation and inhibition were different. It seems that this

analysis mostly quantifies recording noise.

Response: In the whole-cell recordings, the ensemble current is the sum of activities from many single channels. To study the kinetics of the channels and the signaling upstream of the channel opening, power spectra may provide valuable information. This method has been used in vision research to study phototransduction of retinal photoreceptors (Rieke R and Baylor D, 2000, Neuron; Fu YB et al., 2008, Nat. Neurosci; Luo DG et al., 2011, Science). However, this analysis only reveals the integrated kinetics of the entire signaling pathway. The recording noise is subtracted, and only the biological signals are analyzed.

3) The demonstration of competitive antagonism in figure 3e is very nice and a strength of the paper. Does this finding generalize to other inhibitory odors? Or are there different modes of action for the effects of different inhibitory odorants?

Response: We observed similar effects with another inhibitory odor 3-methylbutanol (see Fig. R2), and also on excitatory responses to acetophenone by an inhibitory odor geraniol in Or10a-OSNs. However, we did not examine other odor combinations to rule out other possible modes of actions.

Fig. R2. Competitive inhibition by 3-methylbutanol. (A) A family of superimposed responses of an Or85a-OSN to E-3 (35 ms) in the absence (left) and presence (right) of 10 mM inhibitory odor 3-methylbutanol. (B) Dose-response relationship. 10 mM 3-methylbutanol decreases the $K_{1/2}$ of E-3 responses from 0.3 mM to 8 mM. Collective data: 10 mM 3-methylbutanol decreases the $K_{1/2}$ from 0.8 ± 0.19 mM to 9.0 ± 1.1 mM ($n = 4$; $P = 0.0003$).

4) I wonder if it would clarify the results to speak about the valence of a glomerulus (which can be either activated or suppressed by different odorants) rather than talking about the actions of specific inhibitory or excitatory odors on behavior. This seems to be a simpler explanation of the behavior than conversion of inhibition into excitation by feedback, as suggested in the discussion.

Response: Thank you for the suggestion. We like the idea of the valence of a glomerulus, which is a simpler explanation to our results. To promote future mechanistic studies, we would like to suggest some potential mechanisms of inhibition signaling in olfactory circuits. Therefore, we kept the original discussion.

5) The presentation of the model is very brief and therefore opaque. ...I think it would be helpful to reader if the authors either presented a fuller picture of the full model

in the results, including what assumptions it makes, or simplified the model so it captures only the most important intuitions it makes, or simplified the model so it captures only the most important intuitions. The full model would potentially be better suited for a separate paper in another journal.

Response: According to the suggestions by you and Reviewer #1, we expanded the model part (see Text in RED at Page 13-15).

Minor comments:

Figure 2e is difficult to see at its current scale. Figure 3a might be easier to parse if the different panels were at similar scale.

Response: Done.

Figure 1a looks as though the acetophenone responses is saturating at ~6-8 pA, which is much less than the average basal current of -18 pA. Perhaps it would be useful to report or show the distribution of basal current amplitudes?

Response: Following your suggestion, we reported the distribution of basal inward current amplitudes with the mean \pm S.D. instead of the mean \pm S.E.M. in Line 98-99.

Reviewer #3

We thank you for complementing our study as “interesting”, “fundamental”, “impressive”, and “interest a wide range of readers”. In addition, we are grateful to you for your careful reading and suggestions.

Major concerns:

1) As detailed below, most results are reported as single examples, and no results are supported by necessary descriptions of variance and statistical significance. The authors need to check each figure panel and provide appropriate statistical support for each claim they make.

Response: We obtained reproducible recordings, and the representative data were shown. Following your suggestions, we added the statistics accordingly.

2) The authors used a new procedure to deliver odor stimuli in the aqueous phase while recording from OSNs in dissected antennae. As detailed below, their results show levels of spontaneous activity substantially different from those reported by other groups recording from the same neurons. This disparity is not mentioned by the authors. The potential concern here is the authors’ new preparation may not be functioning in a normal, physiological fashion. This concern is amplified by comparisons the authors make between results obtained through their new procedure in isolated antennal preparations and behavioral results obtained from intact animals. The authors need to directly address this concern: does their new preparation function in a normal, physiological fashion, and, if its properties are somehow different from those of intact animals, how can the authors compare results obtained this way to results obtained from intact animals?

Response: Thank you for pointing out this important concern.

We obtained similar results to those by Carlson’s group with intact animals, in terms of odor-evoked excitation or inhibition for the same ORs. The spontaneous activity rates differ between our recordings and Carlson’s recordings, but the comparison of these absolute rates may not be appropriate because Carlson’s data were obtained from the “empty neurons”, but not from the neurons that endogenously express the corresponding ORs. Generally, different OSNs in *Drosophila* have different intrinsic electrical properties, for example, different spike amplitudes were used to sort out different OSNs in the air-phase SSRs. In addition, the amount of OR expression may differ between our rescue experiments on the OSNs and Carlson’s empty-neuron experiment, which produces different levels of OR-elicited spontaneous activity. For a direct comparison, we performed SSRs from the intact animal and cell-attached recordings in the antennal slice and found similar spontaneous rates of Or85a-OSNs (see Fig. R1). Our calcium imaging from the antennal lobe of live flies where the antenna are stimulated by air-phase delivery of odorants qualitatively validated our aqueous stimulation results (see Text Fig. 6).

Minor concerns:

- 1) The writing is generally very clear, but the manuscript would benefit from light editing to correct small grammatical errors and ambiguities.

Response: Thanks. We polished the writing.

- 2) Title and elsewhere: “Inhibition”: Throughout the text, the authors argue that inhibition carries information, just like excitation. However, in most places, the inhibition that matters here is inhibition in a narrower meaning: reduction of basal firing rate. The authors might want to distinguish this kind of inhibition from the hyperpolarization that can be observed in intracellular recordings.

Response: The inhibition of basal firing is produced by odor-induced hyperpolarization of the OSNs, as shown in our intracellular recordings under a current-clamped configuration (see Fig. R3).

Fig. R3. Acetophenone hyperpolarizes the Or85a-OSNs. Under current-clamped configuration, the patch-clamp recording on an Or85a OSN of an antennal slice showed that 10 mM acetophenone hyperpolarizes the cell and inhibits its basal firing of action potentials.

- 3) Line 46-7 and elsewhere: “a dual code”: This phrasing is misleading because it

suggests the olfactory system is using two separate mechanisms to encode information rather than a single mechanism that employs bidirectional changes in firing rate.

Response: Following your suggestions, we changed the phrase “dual” to “bidirectional”.

- 4) Line 58: “independent sensory code”: again, the authors provide no evidence that inhibition and excitation are evaluated downstream by independent mechanisms.

Response: We deleted the “independent”.

- 5) Line 74-5: “Notably, a loss of odor-evoked inhibition can result in a complete switch of olfactory behaviors.” This sentence does not accurately describe the experimental result. In the behavioral experiments, the authors abolished not only odor-evoked inhibition but also spontaneous activity (transmission, more accurately). It’s not established whether abolishing spontaneous activity has its own effect on coding. (It might, for example, if the lack of spontaneous firing effectively raises the firing threshold of follow cells.)

Response: Thanks. Yes, it is possible that spontaneous activity affects the firing threshold of the follow cells. We modified the sentence as “Notably, the blockage of synaptic transmission results in a complete switch of olfactory behaviors.”

- 6) Line 91-2: “Correspondingly, an outward receptor current was induced by acetophenone...” Later the authors show the outward current is actually a reduction of inward currents. The wording used here may confuse some readers.

Response: Thanks. We added “..., thus yielding an outward receptor current.” to the end of line 127.

- 7) Figure 1a, bottom: What is the rationale for holding the membrane potential at – 80 mV? Does the spontaneous spike rate measured in whole cell mode match the spike rate recorded with sensillum recordings?

Response: Holding the membrane potential at – 80 mV is to increase the driving force of the receptor current, a common practice in patch-clamp recordings. Spontaneous spike rate was not determined in whole-cell mode because of the resting potential is easily changed by the recording quality.

- 8) Figure 1b: Is this figure from a single representative cell? The authors need to establish the reliability of their results by showing statistical support for their claims.

Response: We added the statistics.

- 9) Line 108: typo: Supplementary Fig. 1c,e” should be “Fig.1c and Fig. S1e”.

Response: Thanks. Corrected.

- 10) Line 116-118: “Alternatively, acetophenone may interact with Or85a receptor to close ion channels that are constitutively open and have a reversal potential above

the resting potential of OSNs.” But this possibility has already been excluded by experiments with the Orco mutant, which are described above. Further, do the authors mean to argue that the ion channels by themselves are constitutively active or that the constitutive activities of ion channels are conferred by the constitutive activity of Or85a? The experiments described below do not clearly differentiate these two possibilities.

Response: Yes, the constitutive activities of ion channels are conferred by the constitutive activity of Or85a, which disappeared in our recordings of Or85a-OSNs of the Or85a-knockout flies.

11) Line 138: Fig. 2b,c: the authors should show examples from multiple cells.

Response: We added the statistics.

12) Line 145: “mirrored”: this word is not appropriate; it means that, if one of the curves were reflected along the x-axis then the two curves would be superimposable, and they are not. How about something like: “exhibited similar, although opposite in direction, current-voltage relationships”.

Response: Thanks. We changed this phrase accordingly.

13) Line 146: Fig. 2d: the authors should show examples from multiple cells. Also, data points shown on the I-V curve should be color-coded to indicate whether they were evoked by excitatory or inhibitory odorants.

Response: Thanks. We color-coded the I-V curves and added the statistics.

14) Line 154: Fig. 2e: Although this effect looks consistent across cells the authors should quantify the variability with standard statistics: show a p-value.

Response: Done.

15) Line 164: Figs. 3a: These measurements need to be quantified across cells with appropriate statistics.

Response: Done.

16) Line 166: Fig. 3b: These measurements need to be quantified across cells with appropriate statistics.

Response: Done.

17) Line 168: Fig. 3c: These measurements need to be quantified across cells with appropriate statistics.

Response: Done.

18) Line 169: Fig. 3d: These measurements need to be quantified across cells with appropriated statistics.

Response: Done.

19) Line 175: “significantly”: the authors should not use this word unless it is accompanied by statistic validation, which is not proved here.

Response: We added the statistics in the Figure legend (Line 955-956).

20) Line 176: Fig. 3e: The authors show some numbers in Table 1, but the appropriate statistics must be calculated and shown in the text or figure caption.

Response: Thanks. We added it to the text.

21) Line 191: “cell-attached recordings”: It would be preferable to replace the cell-attached results with those from sensillum recordings, a technique allowing for more naturalistic experimental conditions. This is especially important because the spontaneous spike rates reported here do not match published sensillum recordings from John Carlson’s group (spontaneous spike rate sensillum recordings (Hallem et al., 2006, Kreher et al., 2008): Or82a: 16 Hz; Or10a: 14 Hz; Or42b: 7 Hz; Or43a: 21 Hz; Or85a: 14 Hz). The authors need to address these disparities in spontaneous activity rates.

Response: The comparison of spontaneous rates may not be appropriate (see Response to Major Concern 2 by Reviewer #3). Thus, we avoided to make comparison between the absolute rates. Following your suggestions, we performed both SSRs and cell-attached recordings on the Or85a-OSNs of WT flies. The spontaneous rates were similar between the SSRs and cell-attached recordings (see Fig. R1).

22) Line 246: “Similar results were obtained by disrupting the signaling of odor-evoked inhibition in Or10a-OSNs.” But for all results shown in Fig. 6, transmission or activity of ORNs were silenced. As noted above, it is unclear whether the observed behavioral effects were caused by abolishing the odor-evoked inhibition of basal activity, or by abolishing basal activity itself. It is unclear how the basal spiking of ORNs affects the fly’s ability to follow odor presentation. The authors should discuss this caveat.

Response: Following your suggestion, we added “However, one caveat of these experiments is that, in addition to disrupting odor-evoked OSN inhibition, these experimental manipulations also eliminate the basal firing of OSNs, which may affect the sensitivity of the downstream olfactory neurons³.” at Page 13, Line 262-265.

23) Line 291: “Third, genetic disruption of odor-evoked inhibition...” Strictly speaking, synaptic transmission was disrupted, including spontaneous transmission, not only inhibition.

Response: See Response 22 above.

24) Line 328-9: “hence effectively doubling the number of ORs because the two OR modes can drive opposing behaviors.” Because it has not been established that inhibition of baseline activity provides the same information capacity as excitatory responses, “doubling the number of Ors” is not an accurate comparison.

Response: Thanks. We changed “doubling” to “increasing”.

25) Line 341-2: “Although spontaneous activity is known to be important in early neural development, its widespread existence in adults has remained puzzling.” This issue was investigated by Joseph et al 2012; these authors suggest spontaneous activity is an inevitable consequence of the extreme sensitivity of the OSNs to odors, and that any modification to reduce the spontaneous activity might also reduce sensitivity.

Response: Thanks. We added this reference.

REVIEWERS' COMMENTS:

Reviewer #1 (Remarks to the Author):

I found revisions to be satisfactory and I have no further critiques.

Reviewer #2 (Remarks to the Author):

The manuscript has been revised, in particular to incorporate additional details about the model into the results. I think the manuscript is acceptable for publication with minor revisions:

1) Figure 2 strives to demonstrate that odor-evoked excitation and inhibition are mediated by the same channels. Although I think these are the right experiments to address this question I think the results do not definitively demonstrate that this is the case. For example, the authors state that "Power spectral analysis revealed similar waveforms between the basal inward current and odor-evoked responses." However, panel c shows different spectra for basal (blue) and excitatory (red) activity. Similarly, in d, the authors state that "inhibitory responses exhibited similar, although opposite in direction, current-voltage relationship." I don't think this is true, as the curves have very different shapes, although they do share a reversal potential. The experiment in e-f is confounded by the fact that basal activity changes in the absence of Ca^{2+} . The authors should qualify their claim by saying that their data are consistent with a model where excitation and inhibition arise from the same channels, although some caveats about that interpretation remain.

2) In the results, line 302, please explain in the main text how the information entropy was calculated. The methods indicate that this was quite an involved estimation, performed at least two different ways. But this is not apparent to a casual reader of the results.

minor comments:

line 232: please remind the reader of the Or85a physiology results here to provide context for the behavioral experiment.

line 317: typo: odor numbers

Reviewer #3 (Remarks to the Author):

The authors have satisfied my concerns. I congratulate them on their interesting and important work.